# Discrete and conserved inflammatory signatures drive thrombosis in different organs after *Salmonella* infection

Marisol Perez-Toledo [1,9], Nonantzin Beristain-Covarrubias [1,9], Jamie Pillaye[1], Ruby R. Persaud[1], Edith Marcial-Juarez[1], Sian E. Jossi[1], Jessica R. Hitchcock[1], Areej Alshayea[1], William M. Channell[1], Niek T. J. Wiersma[1], Rachel E. Lamerton [1], Dean P. Kavanagh [2], Agostina Carestia[3], William G. Horsnell[4,5], Ian R. Henderson[6], Nigel Mackman [7], Andrew R. Clark [8], Craig N. Jenne [3], Julie Rayes [2], Steve P. Watson [2] ✉ & Adam F. Cunningham [1] ✉

Inflammation-induced thrombosis is a common consequence of bacterial infections, such as those caused by *Salmonella* Typhimurium (STm). The presentation of multi-organ thrombosis post-infection that develops and resolves with organ-specific kinetics raises significant challenges for its therapeutic control. Here, we identify specific inflammatory events driving thrombosis in the spleens and livers of STm-infected mice. IFN-γ or platelet expression of C-type lectin-like receptor CLEC-2, key drivers of thrombosis in liver, are dispensable for thrombosis in the spleen. Platelets, monocytes, and neutrophils are identified as core constituents of thrombi in both organs. Depleting either neutrophils or monocytic cells abrogates thrombus formation. Neutrophils and monocytes secrete TNF and blocking TNF diminishes both thrombosis and inflammation, which correlates with reduced endothelial expression of E-selectin and leukocyte infiltration. Moreover, inhibiting tissue factor and P-selectin glycoprotein ligand-1 pathways impairs thrombosis in both spleen and liver. Therefore, we identify organ-specific, and shared mechanisms driving thrombosis within a single infection. This may inform on tailoring treatments towards infection-induced inflammation, and single- or multi-organ thrombosis, based on the clinical need.

A key function of inflammation is to provide a rapid response to infection and restrict the dissemination of the pathogen and its components. However, in severe infections, excessive inflammation can also result in the development of a pro-coagulant state that can drive coagulopathies, with deleterious consequences for the host[1]. Thrombosis and coagulopathies are observed in humans and animal models following localized and severe disseminated bacterial infections, such as those caused by the plague bacillus or *Salmonella*[2–4]. Furthermore, a

[1]Institute of Immunology and Immunotherapy, University of Birmingham, Birmingham, UK. [2]Institute of Cardiovascular Sciences, University of Birmingham, Birmingham, UK. [3]Department of Microbiology, Immunology, and Infectious Diseases, Cumming School of Medicine, University of Calgary, Calgary, Canada. [4]Division of Immunology, Institute of Infectious Disease and Molecular Medicine, University of Cape Town, Cape Town, South Africa. [5]Medical Research Council Centre for Medical Mycology, University of Exeter, Exeter, UK. [6]Institute for Molecular Bioscience, University of Queensland, Brisbane, Australia. [7]UNC Blood Research Center, Department of Medicine, University of North Carolina at Chapel Hill, Chapel Hill, USA. [8]Institute of Inflammation and Ageing, University of Birmingham, Birmingham, UK. [9]These authors contributed equally: Marisol Perez-Toledo, Nonantzin Beristain-Covarrubias. ✉ e-mail: s.p.watson@bham.ac.uk; a.f.cunningham@bham.ac.uk

recent study showed that patients with newly diagnosed non-typhoidal salmonellosis (NTS) have an increased risk of developing deep vein thrombosis (DVT) and pulmonary embolism (PE)[5]. In patients with typhoid fever, the risk of thrombosis is coincident with thrombocytopenia, high D-dimer levels, increased circulating fibrinogen levels, disseminated intravascular coagulation (DIC), and an increase in markers associated with endothelium activation, such as soluble von Willebrand Factor and endocan[6]. Importantly, such changes in thrombosis risk and coagulation markers do not correlate with the level of *Salmonella* in the bloodstream as patients have a median detectable burden of only 1 bacterium per ml of blood[7]. This suggests that processes activated because of the infection are the main drivers of the changes in coagulation parameters. However, the mechanisms that contribute to thrombosis development after infections such as those caused by *Salmonella* are less clear.

We have previously reported that after *Salmonella* Typhimurium (STm) infection in the mouse, thrombi form in the liver from seven days after infection and persist for weeks thereafter[8]. Nevertheless, thrombi also form in the spleen within the first day of infection[9], but in this organ, thrombosis resolves rapidly over the next day. Thus, one infection can drive thrombosis in multiple organs with predictable kinetics. In both organs, there is an absolute requirement for clodronate-liposome-sensitive cells (monocyte-lineage cells) for thrombosis. This suggests that the same mechanisms may be shared between both these sites, particularly a need for the cytokine IFN-γ and platelet activation through the podoplanin/C-type lectin-like receptor 2 (CLEC-2) axis, which are key mediators of this process in the liver[8]. However, the discrete kinetics of thrombosis in the spleen suggest that there might also be organ-specific factors that influence this process too, or indeed a core mechanism whereby some factors are shared but others contribute to thrombosis locally in an organ-specific manner.

To assess this, we have performed a side-by-side analysis of which mechanisms are conserved in the spleen and liver at times when thrombosis is established in each organ. This analysis reveals a common mechanism that is shared between these organs but that there is also redundancy for some factors between these sites. This suggests that local factors modulate the risk of thrombus development, and this may need to be considered in future strategies to target infection-induced thrombosis.

## Results

### STm-induced thrombosis in the spleen does not require CLEC-2 or IFN-γ

Our previous work showed that thrombosis in the spleen and liver follows different kinetics. Thrombosis in the spleen is detected 24 h post-infection, whereas in the liver, it takes up to 7 days to develop[8,9]. Moreover, the mechanism driving thrombosis in the liver required IFN-γ and the CLEC-2/podoplanin axis[8]. Therefore, we focused our analysis on day 1 post-infection (p.i.) for the spleen and day 7 p.i. in the liver. To evaluate whether the same factors were needed for thrombosis in the spleen, we infected WT and IFN-γ-deficient mice with STm and evaluated thrombosis 24 h later. IFN-γ-deficient mice had reduced thrombosis in the spleen, but thrombi were still detected in 7 of 10 mice (Fig. 1A, B). In contrast, no thrombi were detected in the livers of IFN-γ-deficient mice (Fig. 1C, D). At this early time point, there was only a minor difference in the bacterial burdens (Supplementary Fig. 1A). In STm-infected *Clec2*<sup>fl/fl</sup>*PF4*<sup>cre</sup> mice, which lack the expression of CLEC-2 on platelets and megakaryocytes, thrombosis levels were comparable to levels observed in WT controls at days 1 and 7 post-infection (Fig. 1E–H). In these mice, the absence of CLEC-2 on platelets did not affect the bacterial control (Supplementary Fig 1B). These results suggest that the mechanism of thrombus formation in the spleen can bypass the need for IFN-γ and CLEC-2.

### Splenic and liver thrombi contain platelets, monocytes, neutrophils, and fibrin

To better understand the mechanisms of thrombosis in both organs, we evaluated the kinetics of thrombi formation prior to 24 h post-infection in the spleen, and prior to 7 days in the liver. After infection, thrombi were detected in the spleen by 8 h and in the liver by 7 days after infection (Fig. 2A). Bacteria detected at 4, 8, and 18 h after infection in the spleen were mostly associated with F4/80<sup>+</sup> macrophages (Supplementary Fig 2A) at sites distal and proximal to vessels. In the liver, on day 1 post-infection, bacteria were found associated with F4/80<sup>+</sup> macrophages, and on day 7, associated with F4/80<sup>+</sup> cells within inflammatory foci (Supplementary Fig 2B). Bacteria were evenly distributed throughout the liver and found at sites proximal and distal to vessels (Supplementary Figs. 2B and 3). We then analyzed the composition of splenic and liver thrombi by immunohistology. We found that neutrophils (Ly6G) and monocytic cells (Ly6C) were present in both splenic and liver thrombi, alongside platelets (CD41) and fibrin (Fig. 2B, C). Neutrophils and Ly6C<sup>+</sup> monocytes were also widely distributed in the red pulp (Supplementary Fig 4). We then analyzed the change in the frequencies of neutrophils and monocytic cells in the spleen and liver after STm infection. In the spleen, the frequency of neutrophils increased from 4 h post-infection (p.i.), with a drastic increase by 18 h (Fig. 2D), which coincided with the detection of thrombi in this organ (Fig. 2). In the liver, compared to non-infected mice, neutrophils were increased from 4 h after infection and remained elevated at day 7 p.i. In contrast, the frequency of monocytic cells increased from day 1 and kept increasing by day 7 (Fig. 2D and Fig. 2F). To better understand the inter-relationship between these cell populations, we performed intra-vital imaging of the spleen and liver at 24 h and 7 days after STm infection, respectively. After infection, clusters of monocytic cells, neutrophils, and platelets were observed interacting in the splenic vasculature at 24 h post-infection (Fig. 2E and Supplementary movies 1–6). In the livers of 7-day infected mice, more neutrophils and monocytic F4/80<sup>+</sup> cells were observed in the liver (Fig. 2F and Supplementary movies 7 and 8). The monocytic F4/80<sup>+</sup> cells were typically more rounded and had a reduced branched morphology than those observed in non-infected mice, consistent with infiltration of the liver by monocyte-derived macrophages. These results suggest that both neutrophils and monocytic cells respond to STm infection and are present in thrombi.

### Thrombosis in the spleen and liver requires neutrophils

Since neutrophils are a major component of thrombi in both the spleen and liver (Fig. 2B, C), we sought to identify whether neutrophils were required for thrombus formation and progression after STm infection. To test this, neutrophils were depleted with a monoclonal anti-Ly6G antibody prior to and during infection with STm (Supplementary Fig 5A), resulting in a reduction in numbers of CD11b<sup>+</sup>Gr1<sup>hi</sup>Ly6C<sup>-</sup> cells in both the spleen and liver (Fig. 3A). Neutrophils in the blood were also 10-fold lower after anti-Ly6G administration (Fig. 3B). Within the timeframes assessed, neutrophil depletion did not markedly affect bacterial control in either the spleen or liver (Supplementary Fig 6A). Nevertheless, thrombosis was completely abolished in both the spleen and liver in the neutrophil-depleted mice (Fig. 3C, D). This suggested that neutrophils are essential for STm-induced thrombosis in both organs. Neutrophil Extracellular Traps (NETs) have been associated with thrombosis development in other models of thrombosis and STm can stimulate NET formation in vitro (Supplementary Fig 7A). Staining of tissue sections for citrullinated histone 3 revealed positive staining associated with both spleen and liver thrombi, consistent with in vivo NET formation (Supplementary Fig 7B). Nevertheless, treating mice with DNAse I 6 h post-infection, before thrombi are detectable in the spleen, did not consistently reduce the level of thrombosis (Supplementary Fig 7C), suggesting NETs may contribute to thrombosis but are not the only mechanism involved. As reported previously[8,9], mice treated with clodronate liposomes, to reduce monocytic cell numbers, also had negligible levels of thrombosis

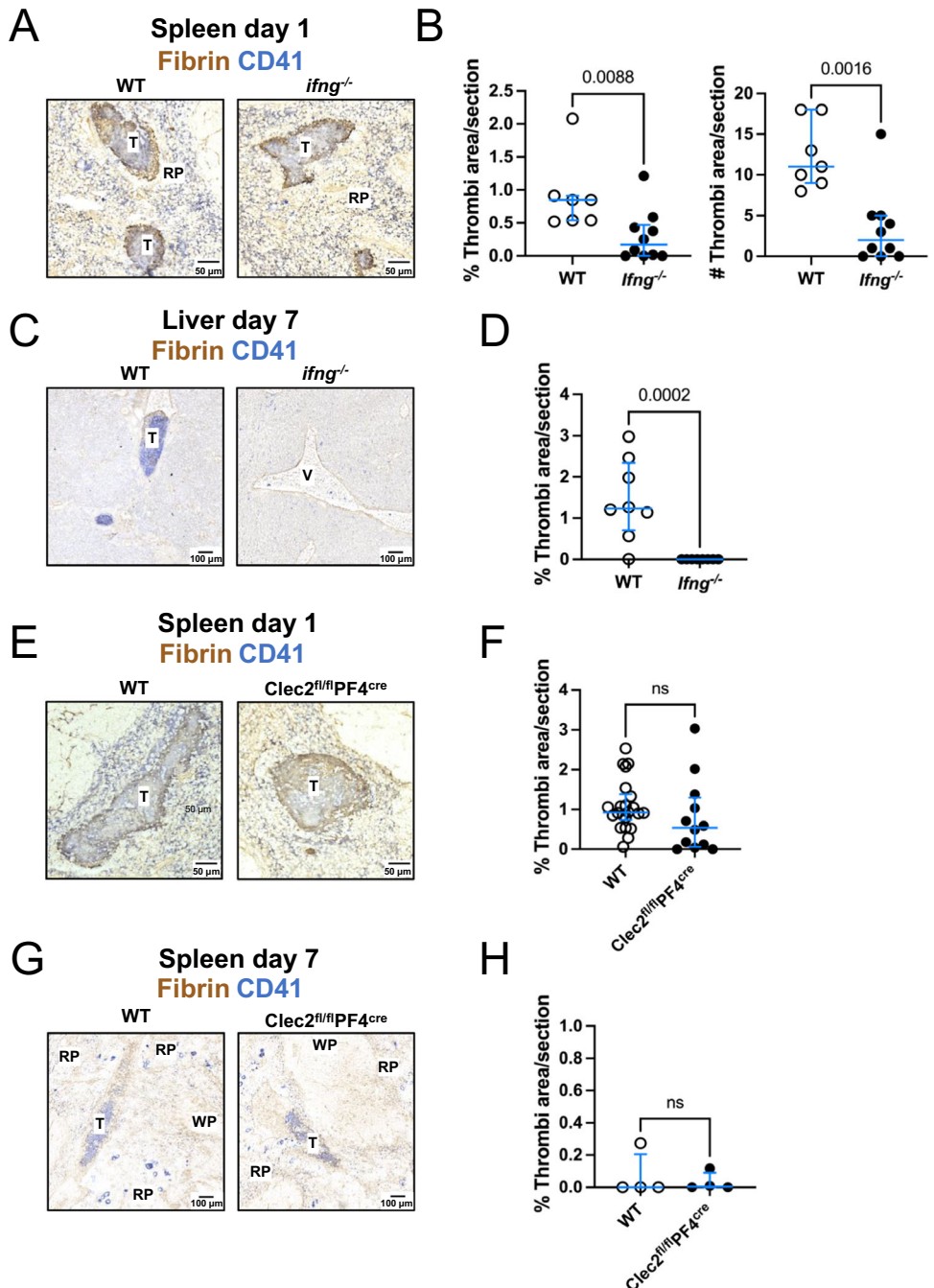

**Fig. 1 | Interferon (IFN)-γ and C-type lectin (CLEC)-2 are dispensable for thrombosis in the spleen after STm infection. A** Representative immunohistochemistry of spleen sections from WT and IFN-γ-deficient mice (*ifng*$^{-/-}$) infected with STm for one day. Brown = Fibrin, Blue=CD41. **B** Percentage of section area (left) and number of thrombi per section (right) in spleens from (A); each point represents data from a single mouse, generated from combining results from two independent experiments for a total of $n = 7$ WT mice and $n = 10$ IFN-γ-deficient mice. **C** Representative immunohistochemistry of liver sections from WT and IFN-γ-deficient mice infected with STm for 7 days. Brown= Fibrin, Blue=CD41. **D** Percentage of section area in livers from (C); each point represents data from a single mouse, generated from combining results from two independent experiments for a total of $n = 8$ WT mice and $n = 8$ IFN-γ-deficient mice. **E** Representative immunohistochemistry of spleen sections from WT and PF4$^{Cre}$CLEC-2$^{fl/fl}$ mice

infected with STm for one day. Brown = Fibrin, Blue = CD41. **F** Percentage of section area in spleens from (**E**); each point represents data from a single mouse, the graph was generated from combining three independent experiments for a total of $n = 22$ WT mice and $n = 12$ PF4$^{Cre}$CLEC-2$^{fl/fl}$ mice. **G** Representative immunohistochemistry of spleen sections from WT and PF4$^{Cre}$CLEC-2$^{fl/fl}$ mice infected with STm for 7 days. (**H**) Percentage of section area in spleens from (**G**); data presented is representative from three independent experiments each with $n = 4$ mice per group and a single point represents data from a single mouse. Median values shown as horizontal lines on each graph. Error bars depict the 75$^{th}$–25$^{th}$ interquartile range (IQR). Each dot represents an independent mouse. Statistical analyses were performed using the two-tailed Mann–Whitney test. Source data are provided as a Source Data file. ns non-significant, T thrombus, V vessel, RP red pulp, WP white pulp.

(Fig. 3E, F; Supplementary Figs. 5 and 6B). Clodronate treatment reduced the number of red pulp macrophages (F4/80$^+$CD11b$^-$) and CD169$^+$ marginal zone macrophages, but it also reduced the number of Ly6C$^+$ monocytes (F4/80$^+$CD11b$^+$Ly6G$^-$Ly6C$^+$) (Supplementary Fig. 8A, B). In the

livers of clodronate-treated and STm-infected mice, Clec4F$^+$ Kupffer cells were not detected (Supplementary Fig 8C). Together, these results show that both neutrophils and monocytic cells are independently required for thrombosis in both spleen and liver.

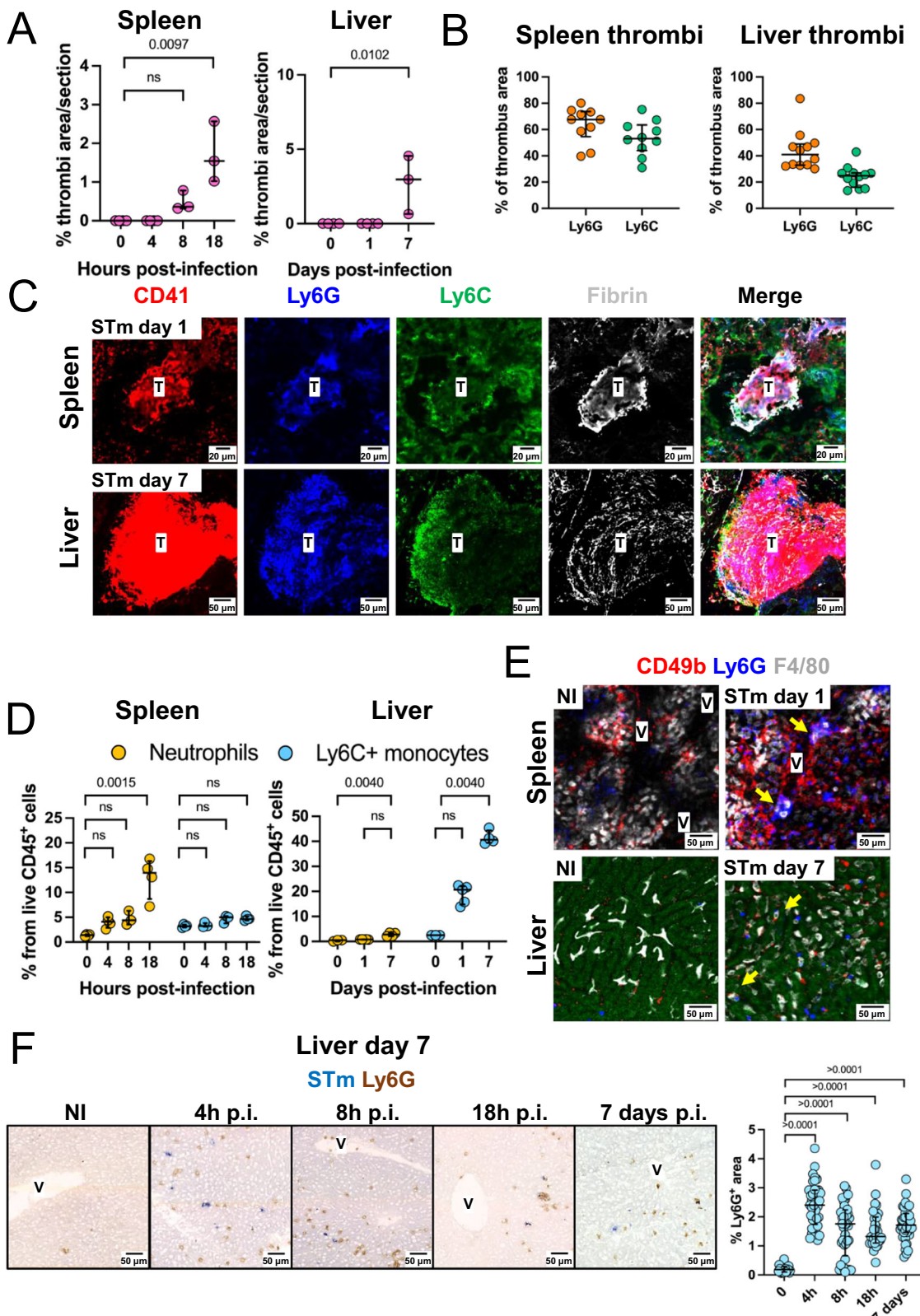

**TNF drives thrombosis in the spleen and liver after STm infection**

The requirement for monocytic cells and neutrophils suggested that inflammatory cytokines contribute to thrombosis. One major cytokine involved in the early response to STm is TNF, which can activate endothelial cells to influence leukocyte migration. After infection, intracellular TNF was readily detected in both neutrophils and Ly6C+ monocytic cells in the spleen and liver after 1 day and 7 days post-infection, respectively (Fig. 4A). Infection of neutrophils and Ly6C+ monocytic cells with STm in vitro also induced TNF (Supplementary Fig 9). At day 1 post-infection, most staining for TNF was associated with F4/80+ cells in the spleen (Fig. 4B). Moreover, clear TNF staining was observed around the blood vessels of livers 7 days after infection (Fig. 4C). To test whether TNF neutralization could prevent thrombosis developing after STm infection, mice were treated with an anti-TNF

**Fig. 2 | STm infection drives the accumulation of neutrophils and Ly6C$^+$ monocytic cells and these cells can be found in thrombi. A** Percentage of tissue area occupied by thrombi in the spleen (0 h $n = 4$, 4 h $n = 4$; 8 h $n = 3$, 18 h $n = 3$) and liver (0 days $n = 4$, 1 day $n = 4$, 7 days $n = 3$) after STm infection across the timepoints indicated. Each point represents data from a single mouse. **B** The percentage thrombus area staining positive for Ly6G (orange) or Ly6C (green), with each point representing individual thrombi from spleens (left, 1 day post-infection) or livers (right; 7 days post-infection). Each dot represents a different thrombus analyzed from $n = 4$ mice per group. **C** Representative images of a spleen thrombus from day 1 (upper row) and a liver thrombus from day 7 post-infection (bottom row). Sections are stained for CD41 (red), Ly6G (blue), Ly6C (green), and fibrin (gray). $n = 4$ mice per group. **D** Flow cytometric analysis of the frequency of neutrophils and Ly6C$^+$ monocytes from live CD45$^+$ cells in spleens (0 h $n = 4$, 4 h $n = 4$; 8 h $n = 3$; 18 h $n = 4$) and livers (0 days $n = 4$, 1 day $n = 5$, 7 days $n = 4$) from mice infected with STm. Representative data from two experiments, each point represents data from a single mouse. **E** Representative fields of view (FOV) of spleen (Upper panel) and liver (bottom panel) obtained by intra-vital microscopy in non-infected mice (NI) and STm infected mice, 1 day (spleen) or 7 days (liver) post-infection. Cells were stained with CD49b (red), Ly6G (blue) and F4/80 (white). Data are representative of two independent experiments each with NI $n = 3$ and STm $n = 4$ mice per group. Yellow arrows indicate the presence of platelets, neutrophils and F4/80$^+$ aggregates. **F** Representative immunohistochemistry of liver sections from non-immunized (NI) or mice infected with STm for the indicated times. The graph shows the quantification of Ly6G$^+$ areas per section generated from F. Each dot represents a different field of view assessed from different time points. Non-immunized (NI) $n = 4$, 4 h $n = 4$; 8 h $n = 3$, 18 h $n = 3$, 7 days $n = 4$ mice. The data are expressed as dot plots, with horizontal lines depicting the medians. Error bars depict the 75$^{th}$–25$^{th}$ interquartile range (IQR). Statistical analyses were performed using the two-tailed Kruskal–Wallis test with Dunn's multiple comparisons tests. Source data are provided as a Source Data file. ns non-significant, T thrombus, V vessel.

antibody and the levels of thrombosis evaluated on days 1 and 7 post-infection (Supplementary Fig 5C). Blocking TNF reduced the development of thrombosis in the spleen and the liver to near undetectable levels (Fig. 4D, E). The administration of anti-TNF antibodies had no impact on bacterial numbers on day 1 post-infection and only a modest impact on day 7 (Supplementary Fig 6C).

It is well-established within the literature that TNF can induce CD62E expression by endothelial cells in vitro[10–14]. In non-infected mice, CD62E was not detectable in the vasculature in either the spleen or the liver (Fig. 5A, B). However, CD62E was upregulated in the spleen and liver after infection at days 1 and 7, respectively. In mice receiving anti-TNF antibodies, less CD62E expression was detected in the vasculature (Fig. 5A–D). To further confirm that the increase of CD62E expression is through TNF signaling, spleen and liver sections from mice deficient in TNF receptors ($p55^{-/-}/p75^{-/-}$) were stained for CD62E (Supplementary Fig. 10). In contrast to WT mice, TNF receptor-deficient mice did not have detectable CD62E staining after infection, nor did they have any detectable thrombi. One consequence of these effects of TNF on the endothelium could be a reduction in the number of immune cells in the tissues. Flow cytometry showed that TNF treatment resulted in a selective reduction in the numbers of neutrophils in the spleen compared to isotype control-treated mice (Fig. 5E), although Ly6C$^+$ monocytic cells increased modestly. In the liver, TNF blocking resulted in reduced numbers of Ly6C+ monocytic cells compared to isotype control-treated mice, but no effect was seen in the numbers of neutrophils (Fig. 5F).

To understand the organ-specific kinetics of TNF and CD62E, spleen and liver sections from non-infected WT mice or mice infected for 4, 8, and 18 h (spleen) or 4, 8, 18 h and 7 days (liver) were stained for both molecules. TNF was detected in the spleen at 4 h post-infection, associated with F4/80+ and Ly6G+ cells (Fig. 6A), whereas CD31+ cell-associated CD62E expression was not detected at 4 h but was detected at 8 h post-infection (Fig. 6B). In the liver, F4/80+ cell-associated TNF was detected at 18 h, but more so at 7 days. The strongest CD31+ cell-associated CD62E expression was only detected in the liver at 7 days post-infection (Fig. 7). These results suggest that TNF promotes thrombosis by inducing CD62E expression in the endothelium that promotes capture of neutrophils and monocytes.

## Tissue factor is essential for thrombosis post-STm infection

An additional effect of TNF on the vasculature is the upregulation of tissue factor (TF) on endothelial cells. Moreover, cells like neutrophils and monocytes can contribute as sources of TF[15]. Because fibrin is a major component of splenic and liver thrombi (Fig. 2B), and fibrin deposition results from the activation of thrombin that can be initiated by TF, we investigated the role of TF in the development of STm-induced thrombosis. Imaging showed that TF was increased in the spleen and liver, 1 day and 7 days post-infection, respectively (Fig. 8A).

TF was detected in the periphery of thrombi, with a similar staining pattern to fibrin (Fig. 8B) and co-stained with Ly6G (Fig. 8C). TF was also detected in perivascular cells, but staining was not associated with CD31$^+$ cells (Fig. 8D). TF promotes the generation of fibrin through thrombin activation. To determine the presence of active thrombin after STm infection, we performed intra-vital microscopy and incorporated a probe that emits fluorescence when cleaved by active thrombin[16]. Elevated thrombin activity was detected in the splenic vasculature on day 1 after STm when compared to non-infected mice (Fig. 8E, F). Similar experiments in the liver on day 7 post-infection showed thrombin activity was increased in the vasculature of STm-infected mice compared to non-infected mice (Fig. 6E, F). Furthermore, active thrombin was detectable in thrombi (Fig. 8G). Finally, the requirement of TF for thrombosis after STm infection was assessed by infecting mice that express low levels of TF (mTF$^{-/-}$, hTF$^{+/+}$) alongside heterozygous control mice (mTF$^{+/-}$, hTF$^{+/-}$)[17]. Thrombi were not detected in either the spleen or the liver of mTF$^{-/-}$, hTF$^{+/+}$ mice (Fig. 8H, I), despite both groups having similar bacterial burdens (Supplementary Fig 6D). These results suggest that upon STm infection, tissue factor, most likely on leukocytes, is required for thrombosis post-STm infection.

## Inhibition of PSGL-1 prevents thrombosis by limiting neutrophil and monocyte recruitment

P-selectin Glycoprotein Ligand-1 (PSGL-1) plays a crucial role in leukocyte recruitment[18,19]. Dual staining for TF and PSGL-1 identified co-localization of these factors within thrombi (Fig. 9A). In models of DVT, peptides that block PSGL-1 function can prevent thrombosis[20]. Therefore, we tested whether blocking PSGL-1 prevented thrombosis after STm infection. We administered an anti-PSGL-1 antibody and evaluated thrombi in the spleen and liver at day 1 and day 7 post-infection, respectively (Supplementary Fig 5D). At these timepoints, fewer thrombi were detected in the spleens and livers of mice receiving anti-PSGL1 (Fig. 9B, C). Blocking PSGL-1 did not affect bacterial control at either of the time points tested (Supplementary Fig 6E). Flow cytometry showed that after PSGL-1 blockade, fewer neutrophils were detected in the spleen 24 h after infection, whilst the frequency of Ly6C$^+$ monocytic cells remained unaffected (Fig. 9D). In contrast, after anti-PSGL-1 blockade, in the liver at day 7 post-infection, fewer CD11b$^+$ Ly6C$^+$ cells were detected compared to the isotype control group, but frequencies of CD11b$^+$Ly6G$^+$ cells were similar between the two groups (Fig. 9E). To visualize the effect of anti-PSGL-1 treatment in the cell populations in the tissues, we stained for CD11b and Ly6G as proxy to identify neutrophils (CD11b$^+$ Ly6G$^+$) and monocytes (CD11b$^+$ Ly6G$^-$). We confirmed that, after treatment with anti-PSGL-1, fewer CD11b$^+$Ly6G$^+$ cells were present in the spleen after 1 day of infection (Fig. 9F), whereas, in the liver, fewer CD11b$^+$ Ly6G$^-$ were detected (Fig. 9G). These results show that PSGL-1 has a role in promoting thrombosis in both the spleen and liver.

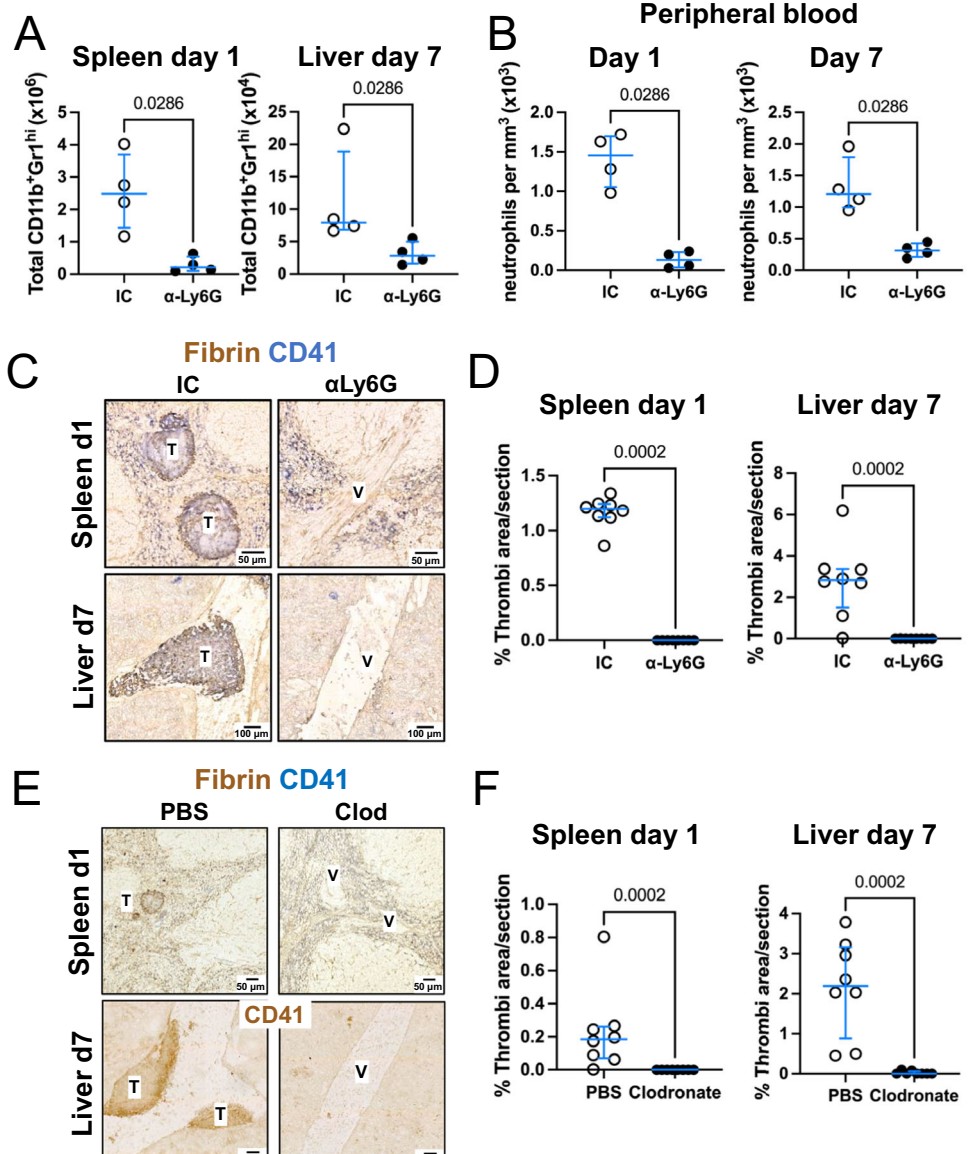

**Fig. 3 | Neutrophils and monocytic cells are required to drive thrombosis in the spleen and liver after STm infection.** **A** Total numbers of CD11b⁺Gr1ʰⁱ cells in isotype control treated (IC) or anti-Ly6G treated mice in the spleen (1 day post-infection) or liver (day 7 post-infection). **B** Numbers of neutrophils per mm³ in the peripheral blood in isotype control (IC) or anti-Ly6G treated mice that were infected with STm for 1 day (left) or 7 days (right). Representative data from two independent experiments with $n = 4$ mice per group. **C** Representative immuno-histochemistry sections from spleens (top row; day 1 post-infection) and livers (bottom row; day 7 post-infection) from isotype control (IC) or anti-Ly6G treated mice (Fibrin= brown, CD41=blue). **D** Quantification of the area occupied by thrombi in spleens and livers from mice represented in (**C**). Data presented are from two

independent experiments combined for a total of $n = 8$ mice per group. **E** Representative immunohistochemistry of spleens (day 1 post-infection) and livers (day 7 post-infection) from mice treated with PBS liposomes or clodronate liposomes (Fibrin= brown, CD41=blue). **F** Quantification of the area occupied by thrombi in the spleens and livers of mice represented in (**E**). Data presented are from two independent experiments combined for a total of $n = 8$ mice per group. The data are expressed as dot plots, with each point representing data from a single mouse, and horizontal lines depict the medians. Error bars depict the 75th–25th interquartile range (IQR). Statistical analyses were performed using the two-tailed Mann–Whitney test. Source data are provided as a Source Data file. V Vessel, T Thrombus.

## Discussion

A potential consequence of serious infections is the development of thrombosis. In this work, we have shown that thrombosis in the mouse model of *Salmonella* infection contains a common pathway that also has organ-specific modulating factors. The common pathway between the organs requires neutrophils, monocytic cells, TNF and TF activity, whereas the most prominent difference between these organs is the differential requirement for IFN-γ and CLEC-2. This suggests discrete thrombogenic pathways that are organ-specific, but that converge in a common mechanism involving inflammation and coagulation.

*Salmonella*-induced splenic and liver thrombi contain platelets, neutrophils, monocytes, and fibrin. The importance of neutrophils and monocytes is demonstrated by the finding that both cell types are needed to support thrombus formation. Additionally, intravital microscopy identified the close interactions between both leukocyte subsets, platelets, and the blood vessels. In the spleen, cellular aggregates consistent with thrombi morphology were observed and had a similar presentation as thrombi imaged using histology. Although technical limitations restricted our study of the portal vein in the liver, where thrombi are mostly found, interactions between neutrophils and monocytic cells in the sinusoids were observed and suggests that

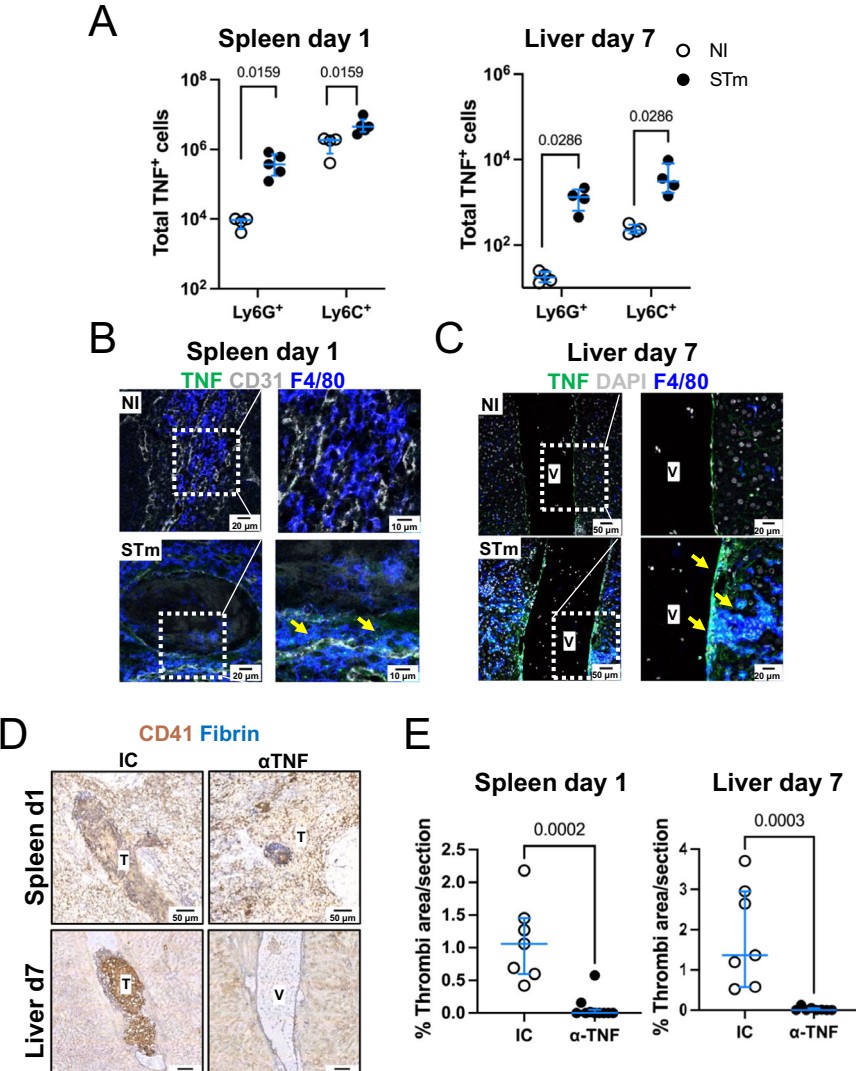

**Fig. 4 | Blocking TNF prevents thrombosis after STm infection. A** Total numbers of TNF-producing neutrophils (CD11b⁺Ly6G⁺) and Ly6C⁺ monocytes (CD11b⁺ Ly6C⁺) determined by flow cytometry in the spleens (NI $n = 4$, STm $n = 5$) and livers (NI $n = 4$, STm $n = 4$) of mice infected for 1 or 7 days respectively. Representative data shown from at least two independent experiments. **B** Representative images of spleen sections from mice infected for 1 day stained for TNF (green), CD31 (gray) and F4/80 (blue); $n = 4$ mice per group. **C** Representative images of liver sections from mice infected for 7 days stained for TNF (green), DAPI (gray) and F4/80 (blue); $n = 4$ mice per group. In (**B**) and (**C**) the right-hand image corresponds to a higher magnification of the area within the dotted white square and the yellow arrows indicate sites of positive TNF staining. **D** Representative immunohistochemistry of spleen and liver sections from mice infected for 1 or 7 days, respectively, and treated with either isotype control antibody (IC) or an anti-TNF blocking antibody (α-TNF) (Fibrin=blue, CD41=brown). **E** Quantification of the area occupied by thrombi in spleen and liver from (**D**). Data presented are from two independent experiments combined. Spleen IC $n = 7$, spleen α-TNF $n = 10$, liver IC $n = 7$, liver α-TNF $n = 8$. Each point represents data from an individual mouse and horizontal lines depict the medians. Error bars depict the 75th–25th interquartile range (IQR). Statistical analyses were performed using the two-tailed Mann-Whitney test. Source data are provided as a Source Data file. V Vessel, T Thrombus.

similar interactions can occur in bigger blood vessels. Moreover, there is a co-requirement for neutrophils and monocytes for thrombosis development in both spleen and liver, although there may be subtle differences in their roles depending on the organ. For example, our results indicate that neutrophils play a major role for thrombosis in the spleen, whereas in the liver, monocytes may play a more dominant role. This possibility is supported by the observation that blocking TNF and PSGL-1 reduced thrombosis in both organs but had distinct effects on the numbers of neutrophils and monocytes in either organ. Nevertheless, a recent report showed that clodronate administration also impairs neutrophil function, without affecting cell viability[21]. Hence, we cannot rule out that the results of our study with clodronate are not due in part to a collateral effect on neutrophils.

Considering the above, how the triad of platelets, monocytes and neutrophils combine to promote thrombosis remains unresolved. A possibility is that neutrophils, through the production of NETs, promote coagulation by mechanisms previously described, including capture and presentation of TF[22,23], as well as directly affecting fibrinolysis[24]. Moreover, neutrophils can directly activate platelets through CD62P/PSGL-1 interactions[25]. On the other hand, monocytes can be a source of TF, which is enhanced by platelet interactions[26,27]. Ultimately, all these processes can activate platelets in the local vasculature, and activated platelets can feed back to affect the function of monocytes and neutrophils. Given the differences in organ platelet density and the vascular beds in each organ, further studies are needed to examine these effects in each organ. Additional future studies will help improve our understanding of how locally infiltrating inflammatory neutrophils and monocytes regulate the coagulation system more widely or platelets specifically.

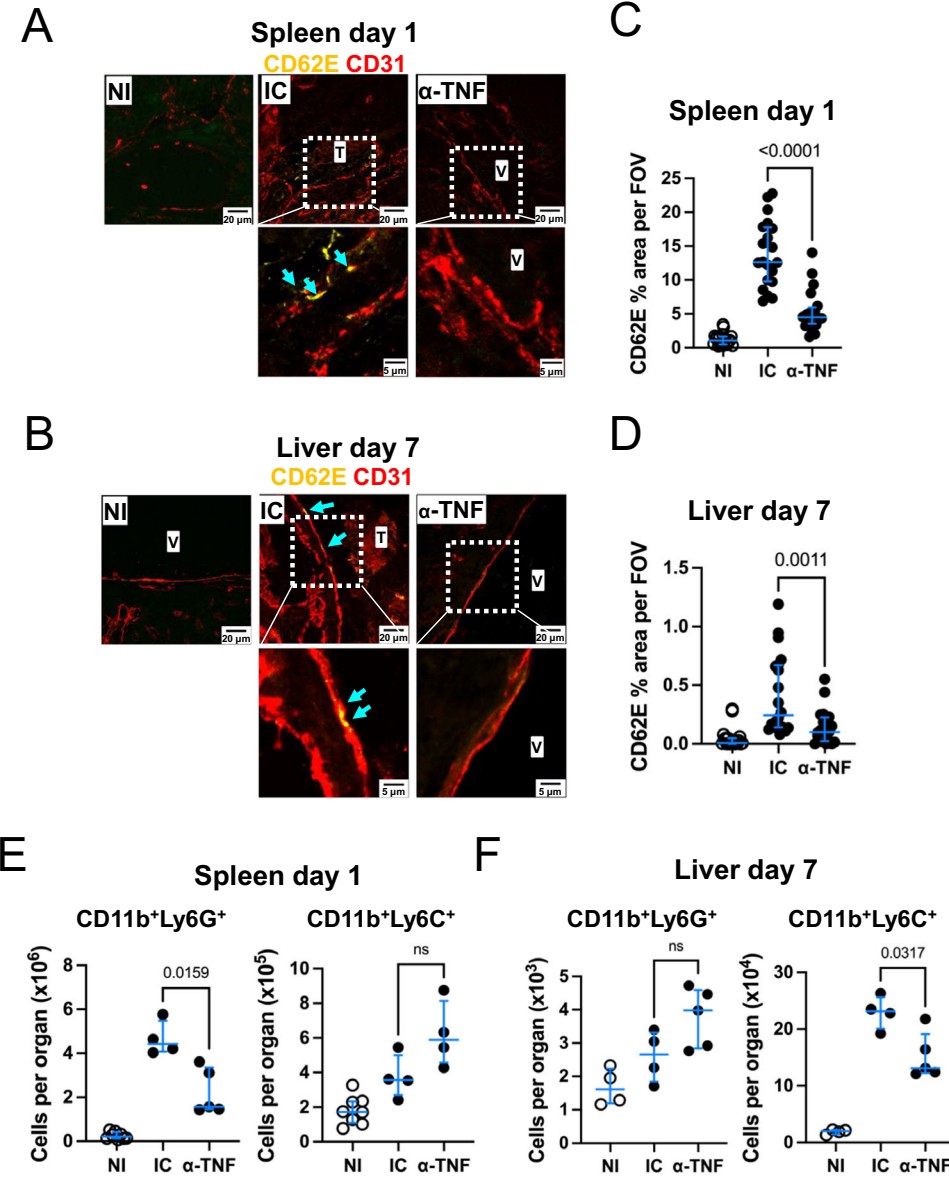

**Fig. 5 | Blocking TNF induced by STm infection reduces CD62E expression.**
**A**, **B** Representative immunofluorescence images of spleen (**A**) and liver (**B**) sections from mice infected for 1 or 7 days, respectively, and treated with either isotype control antibody (IC) or an anti-TNF blocking antibody (αTNF). Sections were stained for CD62E (yellow) and CD31 (red). In each case, images in the bottom panel are a higher magnification of the areas marked by the dotted white squares. NI non-immunized, IC Isotype control, α-TNF anti-TNF antibody. Cyan arrows indicate areas of positive CD62E staining. **C**, **D** The frequency of CD62E+ area per field of view (FOV) in spleen (**C**) and liver (**D**) from sections represented in (**A**) and (**B**) generated from $n = 4$ mice per group and a minimum of 5 FOV per mouse, with

each dot representing a single FOV. Total numbers of CD11b+Ly6G+ cells (neutrophils) and CD11b+Ly6C+ cells (Ly6C+ monocytes) determined by flow cytometry in spleens (**E**) and livers (**F**) of non-immunized (NI), isotype control (IC) or anti-TNF treated mice infected as described for (**A**) and (**B**). Representative data are from two independent experiments. Spleen NI $n = 8$, spleen IC $n = 4$, spleen anti-TNF $n = 5$, liver NI $n = 8$, liver IC $n = 4$, liver anti-TNF $n = 4$. The data are expressed as dot plots, where each dot represents an individual mouse. Horizontal lines depict the medians. Error bars depict the 75th–25th interquartile range (IQR). Statistical analyses were performed using the two-tailed Mann-Whitney test. V Vessel, T Thrombus.

The differing requirements for IFN-γ and CLEC-2 between the liver and spleen were unexpected. Using a model of inferior vena cava ligation, others have shown that mice with a deficiency in CLEC-2 are protected from developing DVT[28]. Our results suggest that not all mechanisms of thrombosis are conserved at all vascular sites throughout the host. Therefore, the mechanisms that are identified in one site may not be generalizable to another vascular site where thrombi form. Other factors could contribute to the redundancy for IFN-γ and CLEC-2 that we have observed for the spleen. For instance, it may relate to the different timings of thrombosis in the different organs. Alternatively, it may reflect the higher density of monocytic-lineage cells and neutrophils present in the spleen at the time of

pathogen encounter and as such there may be a lower threshold for inducing thrombosis in this organ, thus making the podoplanin-CLEC-2 pathway redundant.

Monocytes and neutrophils could promote thrombosis after STm through the production of TNF, which is necessary for thrombosis and is produced by both cell types after STm infection[29]. Whether the contribution of TNF produced by monocytic-lineage cells and neutrophils to thrombosis is similar remains unclear. There may be redundant roles of TNF, produced by each cell type, reflecting the overall importance of this cytokine, or distinct roles when produced by one cell type rather than the other. Future studies will need to address this alongside the overall contribution of

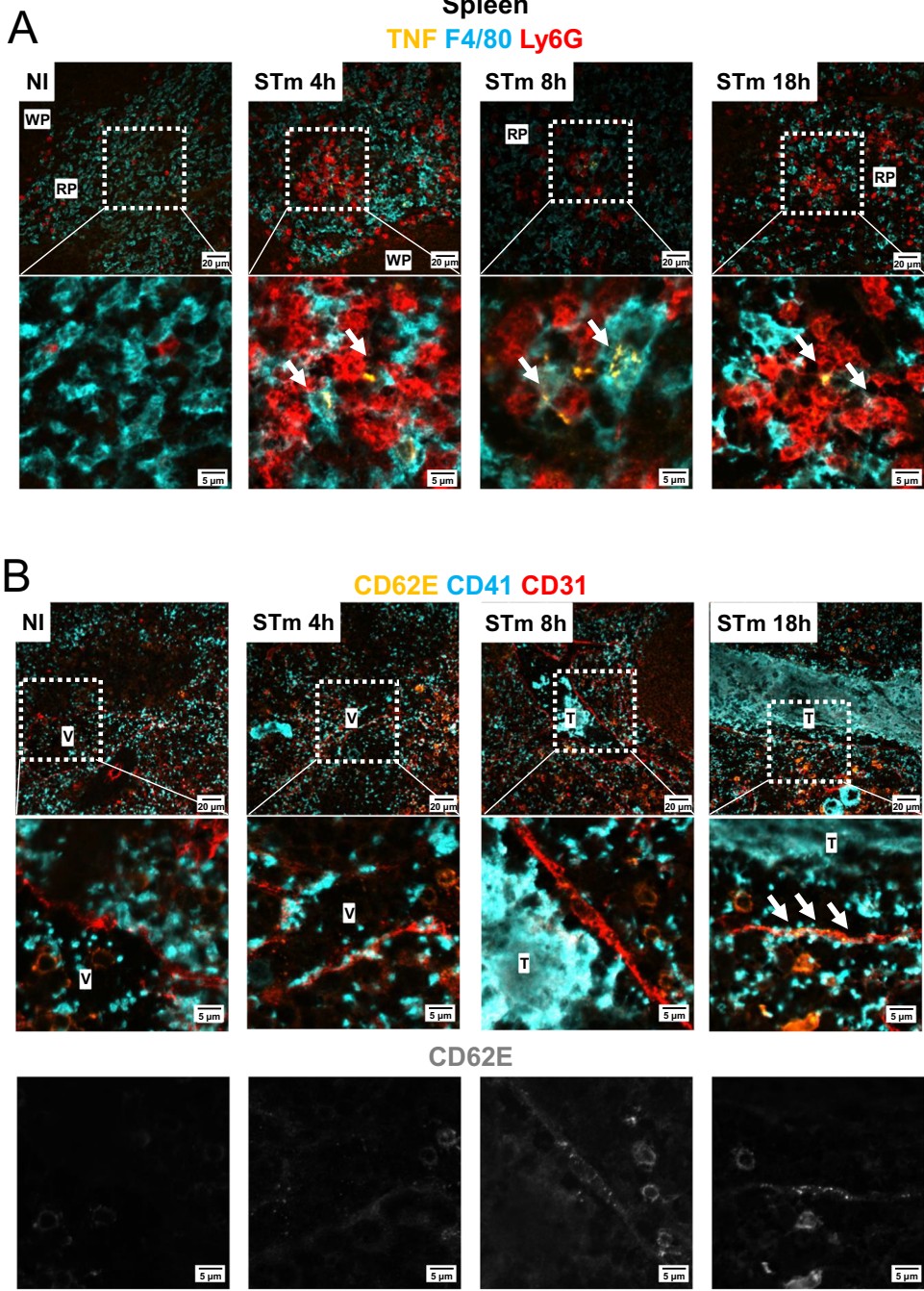

**Fig. 6 | Longitudinal expression of TNF and CD62E in the spleen after STm infection. A** Representative images of spleen sections from mice infected with $5 \times 10^5$ CFU STm SL3261 for 4 h, 8 h or 18 h. Sections were stained for TNF (yellow), Ly6G (blue) and F4/80 (red). The bottom row shows the area within the dotted white squares at a higher magnification. White arrows show positive TNF staining. **B** Spleen sections from (**A**) were stained for CD62E (yellow), CD41 (blue), CD31 (red). The central row shows the area within the dotted white squares at a higher magnification. The bottom row shows CD62E staining only (gray) from the above image. Non-immunized (NI) $n = 4$, 4 h $n = 4$; 8 h $n = 3$, 18 h $n = 3$ mice. NI Non-immunized, V Vessel, T Thrombus.

monocytes and neutrophils to thrombosis. Blocking TNF had a modest impact on bacterial control, which is consistent with the role of TNF in granuloma formation, function and tissue organization[30]. TNF-blocking agents are widely used to treat multiple inflammatory diseases, including psoriasis, Crohn's disease, and rheumatoid arthritis (Reviewed in ref. 31). Still, anti-TNF therapy in sepsis has only shown a modest increase in the survival[32]. However, an increased risk of infection, such as those caused by *M. tuberculosis*, has been reported previously in patients receiving anti-TNF treatment[33]. Therefore, an agent that can block the pathological role of inflammation, without impacting the effector branch for bacterial control, is needed in pathologies derived from infection-triggered inflammation. Thus, targeting molecules such as PSGL-1 may be promising in such situations. Indeed, blocking PSGL-1 molecules can moderate organ damage in models of sepsis[34–36]. Nevertheless, it should be noted that although PSGL-1-deficient mice orally infected with STm did not have increased bacterial burdens in the spleen, they did have more severe infections in the colon and overall, a modest reduction in survival, suggesting this strategy may not apply to all types of infection[37].

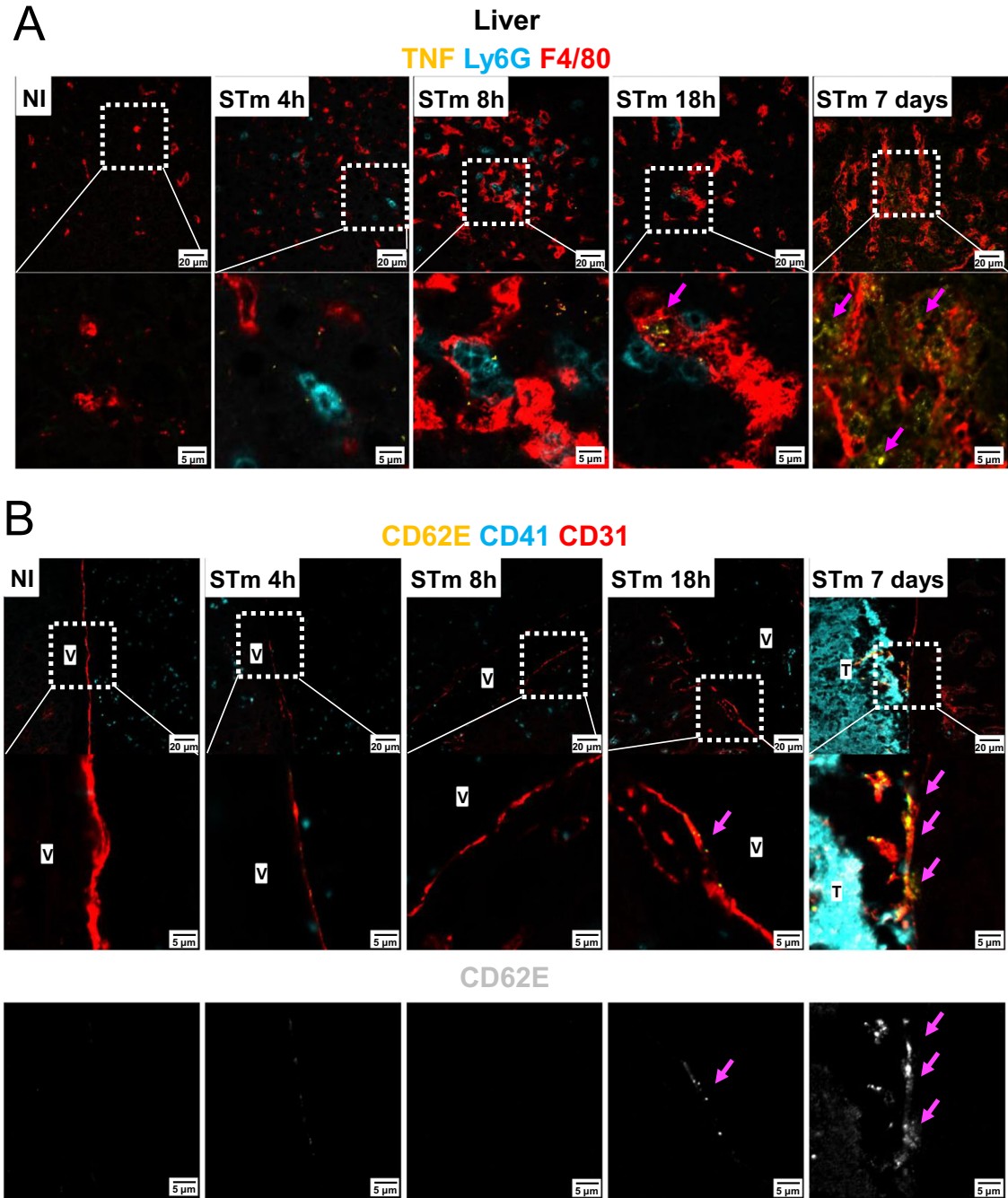

**Fig. 7 | Longitudinal expression of TNF and CD62E in the liver after STm infection. A** Representative images of liver sections from mice infected with $5 \times 10^5$ CFU STm SL3261 for 4 h, 8 h, 18 h or 7 days. Sections were stained for TNF (yellow), Ly6G (blue) and F4/80 (red). The bottom row shows the area within the dotted white squares at a higher magnification. Pink arrows show positive TNF staining. **B** Liver sections from (**A**) were stained for CD62E (yellow), CD41 (blue) and CD31 (red). The central row shows the area within the dotted white squares at a higher magnification. The bottom row shows CD62E staining only (gray) from the above image. Pink arrows indicate positive staining for CD62E. Non-immunized (NI) $n = 4$, 4 h $n = 4$; 8 h $n = 3$, 18 h $n = 3$, 7 days $n = 4$ mice. Representative images from 3-4 mice per group from two independent experiments. NI Non-immunized, V Vessel, T Thrombus.

Infection is an essential step for inducing thrombosis, yet thrombi themselves contain a paucity of bacteria[9] and though bacteria could occasionally be detected at perivascular sites in both the spleen and liver, this was not always associated with the detection of thrombi. The focus of our study has been the host pathways involved in this process, but it is important also to acknowledge that bacterial factors are likely to modulate coagulation. Bacterial type III secretion systems (T3SS) have been documented to have distinct roles during different infections caused by different pathogens including STm and *Yersinia pestis*[38,39]. Thus, T3SS are obvious candidate systems to assess

in the future. Similarly, bacteria are present in both the spleen and liver at similar and near-peak levels in both organs throughout the first week of infection. Furthermore, both neutrophils and monocytic-lineage cells are essential for thrombosis to develop. Despite these parallel features, the distinct kinetics suggest that other factors must contribute to drive thrombosis in specific sites. In the current study, TNF was shown to be a key cytokine involved in this process. Still, it is likely that other cytokines and processes also contribute, potentially before TNF is upregulated. IL-1β and the inflammasome may play some role in this process, as mice deficient in caspase-1 have shown

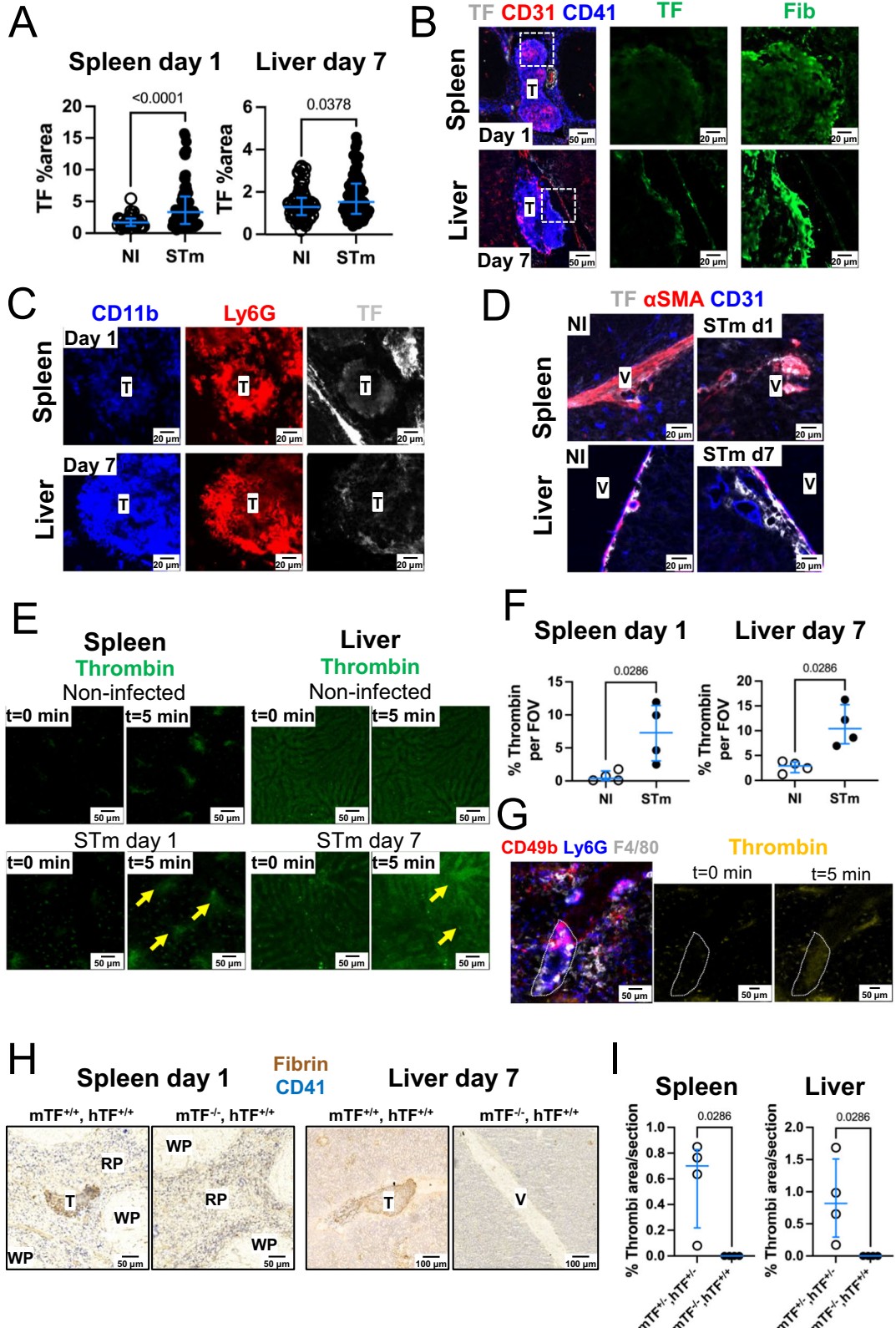

reduced markers of coagulopathy and inflammation after STm infection[38].

One of the contributing factors could be the local activation of the vasculature, which could provide a critical component that "zip codes" where a thrombus will form, with neutrophils and/or monocytic cells present in the blood at the time thrombi form also potentially able to contribute to thrombosis. Therefore, we propose a model where the priming factor is bacterial infiltration into tissues, in which the bacterium resides in cells or niches proximal to the blood vessels. The resulting local secretion of TNF results in the activation of the vasculature and the upregulation of CD62E, which promotes the recruitment of neutrophils or monocytic-lineage cells through the blood, and the interaction of these cells with the vasculature, adjacent to where the bacteria are localized. In the case of the liver, activation of the

**Fig. 8 | STm-induced thrombosis is driven by tissue factor. A** Percentage section area positive for Tissue Factor (TF) in spleens and livers from non-infected controls (NI) or mice infected for 1 day (spleen) or 7 days (liver). Each dot represents a different field of view (FOV) and represents data from a minimum of 10 FOV from each tissue and from a minimum of $n = 4$ mice for each group. **B** Representative images of spleen and liver thrombi (from **A**) stained gray for TF or fibrin (Fib), CD31 (red) and CD41 (blue). **C** Representative images of spleen and liver thrombi (from **A**) stained for TF (gray), Ly6G (red) and CD11b (blue). For (**B, C**) $n = 4$ spleens and $n = 4$ livers from four individual mice were assessed from two independent experiments. **D** Representative images of spleen and liver sections (from **A**) stained for (gray), α-SMA (red) and CD31 (blue); $n = 4$ spleens and $n = 4$ livers from four individual mice were assessed from two independent experiments. **E** Representative FOV of thrombin activation in the spleen (1 day post-STm infection) and liver (7 days post-STm infection) obtained by intra-vital microscopy. $T = 0$ min shows the first frame of the recording before the administration of the thrombin probe. $T = 5$ min shows the frame in the same FOV 5 min after administration of the thrombin probe. Yellow arrows indicate positive thrombin staining. **F** Quantification of thrombin+ regions 5 min after administration of the thrombin probe in non-infected and infected mice. Data presented are from one experiment with $n = 4$ mice per group and are representative of at least two experiments for each organ. **G** Left is a representative image of a splenic thrombus (within the dashed white lines) captured by intra-vital microscopy 24 h after infection, stained for CD49b (red), Ly6G (blue) and F4/80 (gray) next to two images showing thrombin detection (yellow) at $t = 0$ min and $T = 5$ min after the administration of the thrombin probe. Images representative of experiments from four infected mice. **H** Representative sections stained for fibrin (brown) and CD41 (blue) from the spleens (left) and livers (right) of heterozygous mice (expressing human TF and mouse TF, mTF$^{+/-}$, hTF$^{+/-}$) and mice with low tissue expression (mTF$^{-/-}$, hTF$^{+/+}$). Data representative of two independent experiments each with $n = 4$ mice per group. **I** Quantification of the area occupied by thrombi from G ($n = 4$ mice per group). Each point represents data from one mouse, horizontal lines depict the medians. Error bars depict the 75$^{th}$–25$^{th}$ interquartile range (IQR). Statistical analyses were performed using the two-tailed Mann-Whitney test. Source data are provided as a Source Data file. T thrombus, V vessel.

podoplanin/CLEC-2 axis would then contribute to local platelet activation. The exposure and activation of TF would then promote fibrin deposition and eventual thrombus formation (as indicated in the model shown in Supplementary Fig. 11).

## Methods

### Ethics statement
Mice were used in accordance with the Home Office guidelines at the Biomedical Services Unit of the University of Birmingham under the project License P0677946.

### Mice and infection protocol
Male and female C57BL/6J mice aged 6-8 weeks were purchased from Charles River (Strain code 632). Male and female mice were evenly distributed between control and infected groups, except where specifically stated otherwise. Low Tissue Factor mice, IFN-γ-deficient mice[40], PF4$^{Cre}$CLEC-2$^{fl/fl}$ mice[41], and TNF receptor-deficient mice ($pS5^{-/-}p75^{-/-}$)[42] were bred and maintained at the Biomedical Service Unit of the University of Birmingham in specific-pathogen-free conditions. Mice were maintained under a 12-h dark-light cycle, with controlled room temperature between 20 and 24 degrees Celsius, and relative humidity between 40% and 60%. Low Tissue Factor mice were phenotyped as described elsewhere[17]. All mice were euthanized by cardiac bleed under anesthesia, followed by cervical dislocation. The endpoints for all experiments were loss of body weight greater than 20%, or loss of appetite, subdued behavior and piloerection for longer than 48 h. All experiments performed adhered to the endpoints above.

For infection experiments, one day before the experiment, one colony of an attenuated *Salmonella* Typhimurium strain *ΔaroA* SL3261[43] was grown in Luria Bertani (LB) broth (Sigma Cat. No. L3022) at 37 °C overnight. On the day of the experiment, the overnight culture was diluted 1:5 with fresh LB broth and incubated at 37 °C and 180 rpm until the OD600 was near 1. At this point, 1 mL of culture was spun at $6000\,g$ for 5 min and washed with 1 mL of sterile PBS twice. The washed bacterial culture was then diluted in sterile PBS to the desired concentration. Infection was then performed by injecting $5 \times 10^5$ CFU of STm intraperitoneally. In some experiments, monoclonal antibodies were administered during or before the infection to block or deplete specific cell types. Briefly, 500 μg per mouse of an anti-Ly6G antibody (Clone 1A8, Bioxcell Cat. No. BE0075-1), or 300 μg of a blocking anti-TNF antibody (Clone XT3.11, Bioxcell Cat. No. BE0058) were injected i.p. prior to infection, and on day 3 and 5 post-infection (Supplementary Fig. 5A and C). The same amount of rat IgG (Sigma Cat. No. I4131) was used as isotype control. To deplete monocytic cells, 200 μL of clodronate liposomes or PBS liposomes were injected i.p. before infection, and on day 3 and 5 post-infection as described before (Supplementary Fig. 5B)[8,9]. To block PSGL-1, 200 μg of the clone 4RA10

(Bioxcell Cat. No. BE0186) was injected prior infection for the studies in the spleen, and on day 3 and 5 post-infection for studies in the liver (Supplementary Fig. 5D). DNAse treatment was performed by injecting intravenously a single dose of 60 μg per mouse of DNAse I (Roche Cat. No. 4716728001) 6 h post-infection.

To quantify the bacterial burden per organ, approximately 10 mg of spleen and 100 mg of liver were mashed through a 70-μm cell strainer in 1 mL of sterile PBS. 10-fold dilutions were prepared in PBS, and 100 μL of this cell suspension was plated in LB agar plates, and the plates incubated overnight at 37 °C.

### Intra-vital Microscopy
Male C57BL/6 mice were infected as described above. 24 h or 7 days after infection, mice were anesthetized with a mixture of ketamine hydrochloride (Anesketin, Dechra Veterinary Products Inc.) (200 mg/kg) and xylazine (Nerfasin 100, Dechra Veterinary Products Inc.) (10 mg/kg) and liver and spleen imaged as reported elsewhere[22]. Anti-mouse CD49b (1.6 μg, clone HMα2, BioLegend), anti-mouse F4/80 (1.6 μg, clone BM8, BioLegend), anti-Ly6G (1.6 μg, clone 1A8, BioLegend) were injected i.v. 30 min prior imaging. Platelet visualization was performed with an anti-GPIb antibody (Emfret Analytics Cat. No. X649) for some experiments. To visualize active thrombin, mice were prepped as described above, and after imaging was started, a fluorescent probe was injected i.v. (Anaspec Cat. No. AS-72129). 5 to 10-min-long videos were recorded using an inverted Leica SP8 microscope (Leica Microsystems). Image analysis was performed with Fiji (version 2.9.0).

### Immunohistology
5 μm cryosections of spleens and livers were fixed in acetone (Acros Organics Cat. No. 268310025) for 20 min and stored at −20 °C until analysis. Immunohistochemistry (IHC) was performed as described previously[9]. Briefly, sections were re-hydrated in Tris-Buffered Saline pH 7.6 at room temperature and stained in the same buffer with primary antibodies for 45 min. HRP-conjugated or biotin-conjugated secondary antibodies and Vectastain® ABC-AP alkaline phosphatase kit (Vector Laboratories Cat. No. AK-5000) were used as secondaries. HRP activity was detected with SIGMA*FAST* 3-3'Diaminobenzidine tablets (Sigma-Aldrich Cat. No. D4293), whereas alkaline-phosphatase activity was detected using naphtol AS-MX phosphate and fast blue salt with levamisole (All from Sigma-Aldrich). For immunofluorescence, sections were re-hydrated in PBS pH 7.4 and blocked for 10 min with 10% fetal bovine serum (FBS) in PBS. For the liver, additional biotin-blocking steps were performed prior staining with an avidin/biotin blocking kit (Vector Laboratories Cat. No. SP-2001) following the manufacturer's instructions. Antibodies were incubated in the dark at room temperature for 40 min. Slides were mounted in Prolong Diamond (ThermoFisher Cat. No. P36970) and curated during 24 h at

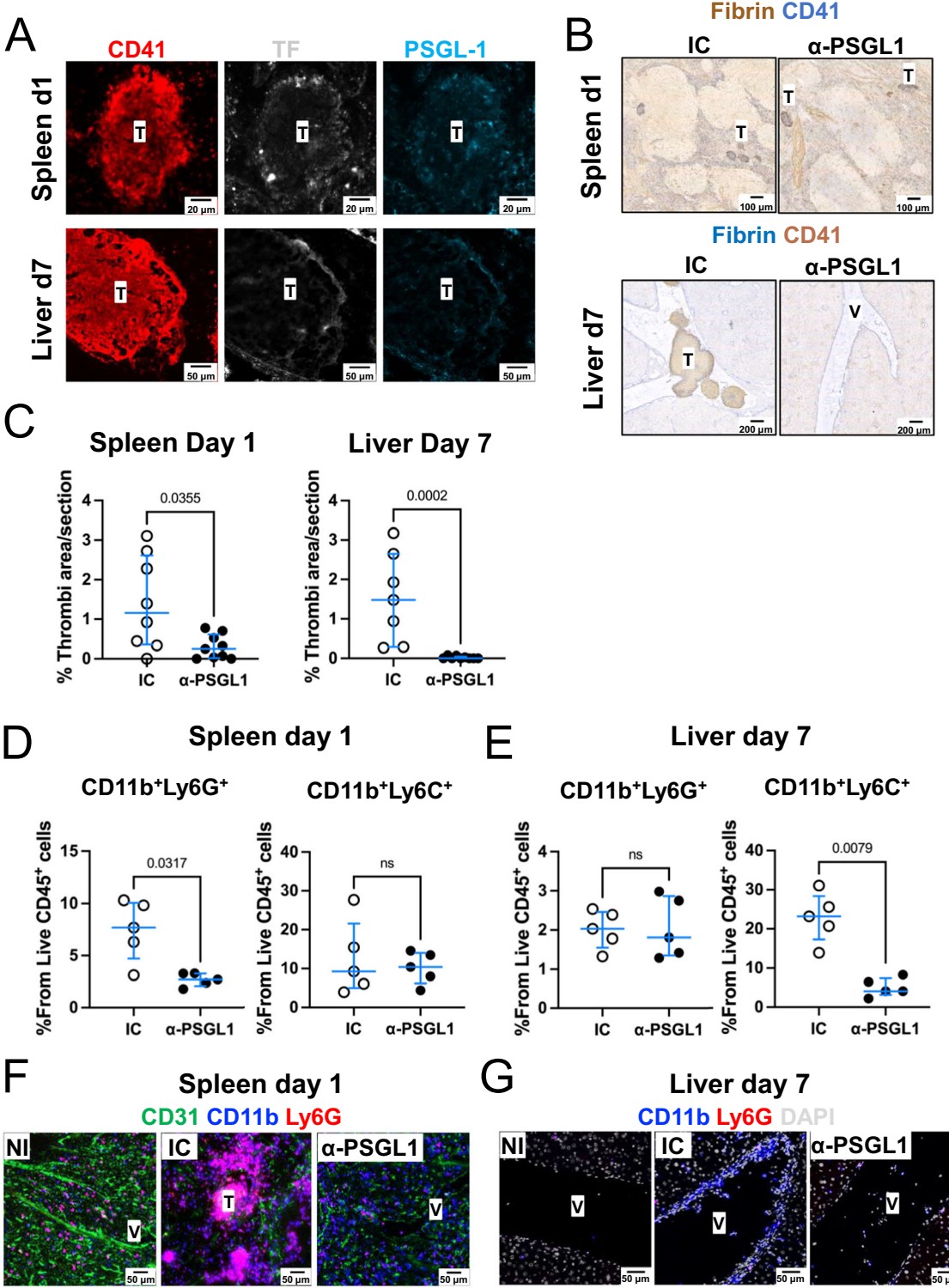

room temperature before imaging. A detailed list of antibodies used can be found in Supplementary Table I

**Flow cytometry**

Single-cell suspensions from spleens were prepared by mashing approximately 20 mg of tissue through a 70 μm cell strainer (Falcon). Red blood cells were lysed with ACK Lysis buffer (Gibco Cat. No. 11509876) and cells resuspended in RPMI with 10% FBS. Liver single-cell suspensions were prepared as described previously[8]. Briefly, whole livers were digested with collagenase IV (Roche, Cat. No. C4-28) and DNAse I (Roche, Cat. No. 4716728001) in a shaking incubator at 37 °C for 20 min.

The suspension was mashed through a 70 μm cell strainer (Falcon). Leukocytes were purified by cell gradient with Ficoll-paque PLUS (Cytiva, Cat. No. GE17-1440). Cells were stained for viability with Zombie Aqua (Biolegend Cat. No. 423101) prior incubation with primary antibodies during 30 min at 4 °C. For intracellular staining of TNF, $5 \times 10^6$ cells were seeded in 48-well plates, stimulated with 5 μg/mL of heat-killed STm, and incubated at 37°C with 5% $CO_2$, in the presence of Golgi Stop (BD Biosciences Cat. No. 554724). After overnight incubation, extracellular staining was performed as described above. Intracellular staining was performed at room temperature using the BD Cytofix/CytopermTM fixation/permeabilization kit (BD Biosciences Cat. No. 554714) according

**Fig. 9 | Targeting PSGL-1 prevents thrombosis and modulates the accumulation of neutrophils and monocytes within organs. A** Representative images of thrombi from spleens (top row) and livers (bottom row) stained for CD41 (red), tissue factor (TF, gray), and PSGL-1 (blue) from mice infected with STm for 1 (spleens) or 7 days (livers). $n = 4$ mice per group, two independent experiments. **B** Representative images from spleen (top row) and liver (bottom row) sections from mice infected with STm (as in **A**) and treated either with isotype control (IC) or anti-PSGL1 (α-PSGL1). Sections are stained for fibrin (brown) and CD41 (blue). **C** Quantification of the area occupied by thrombi in spleens at day 1, or livers at day 7 post-infection (from **B**), from mice treated with isotype control (IC) or anti-PSGL1(α-PSGL1). Data presented are combined from two independent experiments. Spleen IC $n = 8$, spleen anti-PSGL1 $n = 9$, liver IC $n = 7$, liver anti-PSGL1 $n = 9$. **D**, **E** Frequencies of neutrophils (CD11b$^+$Ly6G$^+$) or Ly6C$^+$ monocytes (CD11b$^+$Ly6C$^+$)

quantified by flow cytometry in the spleens (**D**) or livers (**E**) of mice infected as in (**C**) and treated with isotype control (IC) or anti-PSGL1 (α-PSGL1) antibodies. Representative from two independent experiments with $n = 5$ mice per group. **F** Representative images from spleen sections stained for CD31 (green), CD11b (blue), and Ly6G (red) from non-infected mice (NI), isotype control-treated mice (IC) or anti-PSGL-1-treated mice infected for 1 day with STm ($n = 4$ mice per group). **G** Representative images from liver sections stained for Ly6G (red), CD11b (blue), and DAPI (gray) from non-infected mice (NI), isotype control-treated mice (IC) or anti-PSGL-1-treated mice infected for 7 days with STm ($n = 4$ mice per group). In graphs, each point represents the data from one mouse, and horizontal lines show the medians. Error bars depict the 75$^{th}$–25$^{th}$ interquartile range (IQR). Statistical analyses were performed using the two-tailed Mann–Whitney test. Source data are provided as a Source Data file T thrombus, V vessel.

to the manufacturer's instructions. Data acquisition was performed with a CytoFLEX using the CytEXPERT software (Beckman Coulter), and data was analyzed with FlowJo Software v10.10 (BD Life Sciences). Neutrophils were defined as CD11b$^+$Ly6G$^+$, and monocytes were defined as CD11b$^+$Ly6G$^-$CD3$^-$B220$^-$NK1.1$^-$Ly6C$^+$. The proportion of these cells is expressed as frequency from live CD45$^+$ leukocytes. A list of all reagents used can be found in Supplementary Table I.

### In vitro TNF production
Bone marrow cells were obtained from femurs and tibias from C57Bl/6 mice. Monocytes were purified using the monocyte isolation kit (Miltenyi Biotec Cat. No. 130-100-629) following the manufacturer's instructions. Neutrophils were purified by histopaque gradient as described elsewhere[44]. $5 \times 10^5$ enriched monocytes or neutrophils were seeded in a 96-well plate and then incubated with STm SL1344 or STm SL3261 at a multiplicity of infection (MOI) of 1, 5, or 10, in combination with 3 μg/mL of brefeldin A (Invitrogen Cat. No. 15526276). Cells stimulated with media alone were used as negative controls. After 2 h, cells were harvested and stained for flow cytometry analysis as described above.

### In vitro NETs production
Neutrophils were purified by histopaque gradient from C57BL/6 mice as described in the methods section above. Enriched neutrophils were seeded into an 8-well Millicell EZ slide (Merck Millipore Cat. No. PEZGS0816) at a concentration of 10$^5$ cells per well. Cells were stimulated for 4 h with STm SL1344 at a MOI of 10. 100 nM phorbol 12-myristate 13-acetate (PMA, Sigma Cat. No. P8139) was used a positive control. After the incubation, cells were washed with PBS and fixed overnight with paraformaldehyde (PFA) 1%. The following day, cells were incubated with a purified rat anti-mouse Ly6G antibody (clone 1A8), followed by an incubation with a Alexa Fluor 647-conjugated donkey anti-rat antibody (Jackson Immunoresearch). After washing, cells were permeabilized with 0.1% triton x-100 for 10 min, and then stained with a rabbit anti-citrullinated histone 3 antibody (Abcam). Finally, cells were stained with a Cy3-conjugated donkey anti-rabbit (Jackson Immunoresearch) and DAPI. The slides were mounted in prolong diamond, curated for 24 h, and imaged with a Zeiss Axio scan Z1 slide scanner. The frequency of cells producing NETs was calculated by dividing the number of cells that showed DNA protrusions typical of NETs, alongside positive staining for Citrullinated Histone 3, by the total number of Ly6G positive cells, and multiplied by 100.

### Image analysis
The frequency of the area occupied by thrombi was measured using Zen Lite 3.1 (Zeiss), with quantification performed blindly by the researcher. We selected thrombi based on positive staining for CD41 and fibrin in sections stained by IHC. The total area covered by thrombi was calculated by adding up the area of all thrombi in each tissue section and then dividing this by the total area of the section. An example of this calculation can be seen in the diagram provided in

Supplementary Fig. 12A, with representative quantification across serial sections from individual spleens shown in Supplementary Fig. 12B. To quantify C62E or TF positive areas; we selected 5 random fields of view (FOV) per section. Using Fiji[45], each image was converted to 8-bit, and a Gaussian blur and background subtraction were applied. Binary images were obtained after thresholding, and the "measure" function was applied to obtain the total positive pixel area.

### Statistical analysis
The two-tailed Mann–Whitney non-parametric sum of ranks test was used to determine the statistical significance between two experimental groups. For comparisons between more than three groups, the two-tailed one-way ANOVA with Dunnett's multiple comparisons test, or the Kruskal–Wallis test with Dunn's multi-comparison test were performed. The $p$ values were calculated using GraphPad Prism version 10.1.0 and interpreted as significant where the $p$ value was ≤0.05. In all graphs, the data are expressed as dot plots, with horizontal lines depicting the medians. Error bars depict the 75$^{th}$–25$^{th}$ interquartile range (IQR).

### Reporting summary
Further information on research design is available in the Nature Portfolio Reporting Summary linked to this article.

## Data availability
All data are included in the Supplementary Information or available from the authors, as are unique reagents used in this Article. The raw numbers for charts and graphs are available in the Source Data file whenever possible. Source data are provided with this paper.

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

## Acknowledgements

The authors would like to thank the personnel of the Biomedical Service Unit, the University of Birmingham Flow Cytometry Services, and the Microscopy Facilities from the University of Birmingham for their support. The authors would also like to thank Ms. Laura Godin and Dr. Fien von Meijenfeldt for their assistance with experiments. This work was funded by a grant by the Medical Research Council (MR/N023706/1) awarded to A.F.C. R.E.L. was supported by a scholarship awarded by the Wellcome Trust (222389/Z/21/Z). M.P.T. was supported by a Global Mobility Award by the Wellcome Trust. S.P.W. is a British Heart Foundation chair (CH/03/003).

## Author contributions

M.P.T. and N.B.C. conceived and designed the analysis, performed the experiments, and wrote the paper. J.P., R.P., E.M.J., S.E.J., J.R.H., A.A., W.M.C., R.L., A.C., D.K. and N.T.J.W. performed experiments. W.G.H., I.R.H., N.M., C.J., and A.R.C. contributed reagents. J.R., S.P.W., and A.F.C. discussed the results and supervised the project. All authors contributed to the final version of the manuscript.

## Competing interests

The authors declare competing interests.
