## [Transparent Peer Review file · Nature Communications]

Discrete and conserved inflammatory signatures drive thrombosis in different organs after Salmonella infection

Corresponding Author: Professor Adam Cunningham

Version 0:

Reviewer comments:

Reviewer #1

(Remarks to the Author)

This manuscript by Perez-Toledo et al describes an elegant study about inflammatory signatures or factors that drive thrombosis induced by Salmonella Typhimurium (STm) infection in different organs. The authors had previously identified chronological differences between thrombosis in the spleen and in the liver. In this study they found that the clots in both organs contained neutrophils, monocytes, platelets and fibrin. Neutrophils and monocytes were independently required for thrombosis, the former played a more prominent role in the spleen and the latter in the liver. In addition, TNF α , PSGL-1, tissue factor (TF) and thrombin played important roles in thrombosis in both organs as inhibition of TNF α , PSGL-1 and TF suppressed thrombosis. They concluded that STm infection triggered a chain of related inflammatory / prothrombotic events, including TNF α release, up-regulation of endothelial expression of E-selectin, leukocyte infiltration into the organs, increased TF expression and thrombin production, led to thrombosis. They believe that the organ-specific and the broader prothrombotic mechanisms they identified will allow targeted treatments of inflammatory thrombosis induced by systemic infections.

General Comments:

1. Inflammatory/immune multi-organ thrombosis induced by severe viral and bacterial infections is potentially serious. It could lead to multiple organ damage and death. The mechanisms and cellular pathways mediating this type of thrombosis are still not fully elucidated. The novel findings of this elegantly designed study provide further insights into the pathogenesis of infection-induced thrombosis. Their identification of organ-specific cellular processes or pathways may help to explain why thrombosis occurs at a particular time in a specific organ post-infection. Their data may aid development of new, targeted treatments of inflammatory thrombosis resulting from infections.
2. The investigational approach and experimental designs used are appropriate and scientifically sound. The use of unique transgenic mice and specific inhibitory antibodies, add strength to their data. Their conclusions are logical and supported by their findings. The illustration in Figure 8 is helpful to the readers in understanding their proposed model of STm infection-induced thrombosis.
3. The most interesting and significant finding is the chronological differences of thrombosis in the liver and the spleen. The authors attributed this to differences in organ-specific kinetics without identifying them nor providing reasons how they would account for the chronological differences. Identification of these specific kinetics would greatly strengthen the manuscript.
4. The authors hypothesize that bacterial infiltration into the tissues is the priming factor for the thrombosis. They also hypothesize that bacteria, residing in the cells or niches near the blood vessels, prime the immune cells and endothelial cells for thrombosis. However, the authors have not shown that the bacteria actually resided in these cells or niches. This evidence would provide confirmation of their hypothesis.
5. The authors also state that the bacteria were present at similar and near peak levels in the liver and spleen, through the first week of infection, without indicating the location of the bacteria within the organs. Was the distribution of the bacteria evenly spread throughout the organ or was it uneven?
However, it is the presence of bacteria at the "priming" sites adjacent to the blood vessels that is most important for induction of thrombosis. Did the bacterial level at the "priming" site(s) peak at a different time in the spleen (day 1) from that in the liver (day 7)? If so, this could explain the chronological differences. Please discuss.

Specific Comments. Major:

6. Page 7 and Figures 2 and 3. Role of neutrophils and monocytes in STm-induced thrombosis. Using cell depletion studies, the authors showed convincingly that both cell types are required for STm-infection induced thrombosis in the liver and the spleen. Peak level of neutrophils in the spleen at 18hour post-infection coincided with the peak level of thrombosis in this organ, and this is consistent with the prominent role of neutrophil in thrombosis induction. However, in the liver, late rise in neutrophil level peaking at day 7 post-infection is surprising. Neutrophils often respond to acute infection very quickly, usually within first few hours, if not in the first day. Furthermore, in figure 2C, neutrophils at its peak in the liver were only about 3% of CD45+ cells. It seems surprising that neutrophil deletion would have such a profound effect on thrombosis, decreasing the thrombi area per section to zero %. Could the authors explain these surprising findings?

7. It is well known that neutrophils contribute to thrombosis via NETosis. Since the authors showed clear evidence that neutrophils played a critical role in STm infection-induced thrombosis, do they have any data on the contribution of NETosis to STm infection-induced thrombosis?

Minor Comments:

8. The authors need to provide more methodological details about how thrombi areas/section were quantified. Was the counting performed objectively without observer's bias? For example, was counting was done independently by at least two persons? Were the fields of view (VOC) randomly selected to avoid selection bias in the results?

9. Antibody staining in some figures is suboptimal. The staining, particularly the blue stain, is so weak that it is very difficult to see. For examples, in Fig 3C, the blue stain in CD41+ platelets is very faint and hence it is very difficult to see the blue-stained platelets in the IC and α Ly6G pictures. The same with the blue fibrin stain in Fig 4D Liver picture, and Fig 6H Liver mTF+/, hTF+/>.

9. On page 10, lines 217-219. The sentence, "In contrast, in the liver day 7 post-infection fewer CD11b+ Ly6G- cells are detected, but frequency and numbers of CD11b+ Ly6C+ cells were similar between the two group (Fig 7F and &7G)". There is a mistake(s) in this sentence. Firstly, Fig 7G shows in day 7 liver picture, CD11b+ Ly6C+ cells are significantly more than in IC mice than in α -PSGL1 treated mice, not similar in numbers as stated in the above sentence. No CD11b+ Ly6G- mice are shown in either group of mice (Fig 7F and &7G). The authors probably mean CD11b+ Ly6G+ mice. If so, Fig 7G shows the number of CD11b+ Ly6G+ cells are the same in the two groups, not fewer as indicated by the authors in the above sentence.

10. Supplementary figure 4. It would be helpful to show data of Wild Type (control) mice for comparison.

Reviewer #2

(Remarks to the Author)

In this study, Perez-Toledo and colleagues examine the cellular and molecular mechanisms that drive sTm-driven thrombosis formation, and define differences in these processes in the spleen and liver shortly after infection in mice. While both neutrophils and monocytes are found in thrombi at both sites, their relative contributions seems to differ, suggesting distinct mechanisms. The involvement of TNF α , CD62E and PSGL-1 in this process is demonstrated through blockade and KO studies, and collectively show a step-wise inflammatory cascade that regulate thrombus formation. This study suggests that while the basic composition of thrombi are similar in the spleen and liver, there may be avenues for select therapies that target processes that are specific for a particular organ system. This is an interesting and well written study that builds on previous knowledge and model systems established by this group, and further describes thrombi mechanisms at two different organ sites that has may have implications on disease progression in humans. I have the following comments:

1. Abstract, Line 113: "These results suggest that the mechanism of thrombus formation in the spleen is independent of IFN γ and CLEC-2" is not supported by data in Figure 1B, where % thrombi/section is significantly reduced in IFN γ KO mice. What is the definition of "detected in 50% of mice" (Line 107) and how was this determined? This conclusion needs to be revisited: with CLEC-2 expression being the only significant difference between thrombi formation at the two tissue sites, the weight of this statement and main conclusions of this study becomes less novel. Similarly, in Figure 8: the role of IFN γ is ignored in the spleen. What is the rationale to diminish their contribution here?

2. Observations that thrombi formation in spleen after STm infection is unchanged in mice lacking Clec-2 on platelet cells is interesting, but only timepoints up to 24 hours is shown. Can you rule out that the kinetics of thrombus formation is unchanged in the absence of Clec-2 beyond 24 hours during the resolution phase of infection? While thrombus development seems to resolve within a few days in wildtype spleen, it is important to confirm that there is no differences between the groups over time. Do individual thrombus persists throughout the infection or do they form/collapse in waves? Can this data also be presented as % occlusion?

3. Figure 1B/D, 3A/C, 4E, 5C/D, 6 etc: the figure legend indicates that each data point is from a single mouse. I find it odd that a single image from the spleen is taken from each mouse for quantitation. I suggest the authors analyze multiple images from different splenic sections and plot the mean values from each mice for a more robust, representative data analysis. This is especially important for Figure 1 as it lays the foundation for the main rationale of this study.

4. The methods and figure legends do not indicate the sex of mice used in all experiments: please clarify if either or both were used and in which studies stated in the figure legends. If only one sex is used, justification for this is needed and

conclusions reflecting this (eg abstract, discussion). It is indicated that IVM studies were done in males: again, a justification is needed why imaging studies were performed exclusively in males. Please also include animal numbers for each figure panel in the legends.

5. I'm having difficulties interpreting data presented in Figure 2. With both neutrophil and monocytes found in thrombi at both sites, does the increased neutrophil accumulation in spleen (and increase of monocytes in the liver) indicate that there is more of each cell type in each thrombi or are they accumulating in other regions of the organ? To complement the IVM studies, the authors should perform IHC to determine whether discrete regions of the spleen occupy different proportions of the two immune cell types. Can we conclude that increased neutrophil counts in spleen is solely due to increased thrombi formation?

6. TNF α blockade has a drastic effect on thrombus formation in vivo, yet its effect on neutrophil homing and monocytes is either modest or non-significant. How do the authors explain the role of these innate cells here? Do platelet numbers in the blood go down after treatment as an alternative explanation? Similarly in Figure 7E: PSGL-1 blockade seems to decrease neutrophil recruitment into the spleen but not monocytes, whereas the reverse is true in the liver. Does this indicate that thrombus formation is driven by different cell types in these organs?

7. There is higher accumulation of neutrophils in the spleen compared to monocytes in Figure 2C, but higher numbers of TNF α + monocytes in both organs. Please discuss this in the context of their respective contribution to thrombus formation.

Minor comment:

1. Please include IVM from uninfected spleens for studies in Figure 2D.
2. Line 182: I would argue that CD62E expression facilitates leukocyte rolling and retention, rather than migration.
3. Please replace low resolution images with higher resolution ones: Figure 1A/C, Figure 2D, Fig 7D/F etc.
4. Figure 3: some experiments include 4 mice and other show data using 8 mice. For consistency, please include all data in this figure from 2 independent (n=4) data for a total of 8 mice.
5. Figure 6A: please use a different color for the median line, it is not visible when everything is in black.
6. Line 318: please check ketamine dose used (likely 200mg/kg)
7. Please include higher magnification images for Figure 5A/B, 6D to better visualize co-localization between the indicated signals.
8. What are the effects of anti-platelet agents, and would the authors predict that they will clear thrombi in the spleen and liver equally?

Reviewer #3

(Remarks to the Author)

This manuscript aims to address the contribution of systemic Salmonella infection to differences in innate immune cell contribution to organ thrombosis. Overall, the results of neutrophil monocyte differences in the course of infection-mediated thrombosis in liver vs spleen are interesting and the intravital images and movies are wonderful.

Major concerns:

1. It is unclear to this reviewer why COVID19 in this manuscript is equated to the pathophysiology of Salmonella infection. Do the authors mean to imply that thrombotic outcomes as a function of intestinal bacterial infection and lung viral infection are the same? If so then there needs to be a lot more evidence provided for the reader. I would remove any reference that

Salmonella can somehow similarly affect thrombosis as COVID, as it misleads the reader.

2. The use of this particular aroA-negative strain of Salmonella is also puzzling when it comes to translational implications. AroA elimination from Salmonella leads to an overall increase of TNF α from the host when compared to the wt. strain in addition to "increased sensitivity to penicillin, complement, and phagocytic uptake" (PMID:27601574) changing the overall impact and pathophysiology of infection. In addition, the virulence of the strain is attenuated and thus the lack of change in replication in organs is not surprising. Replicating the results with an appropriate virulent (non-modified) Salmonella is of essence for the proper interpretation and generation of these potentially valuable results.
3. Coagulation and immuno-thrombotic formation during bacterial infection serves the purpose of containing and confining the bacteria preventing it from spreading. Does the use of this attenuated strain interfere with the process allowing for the pathogen dissemination to other organs? Does the T3SS play a role for thrombotic outcomes or dissemination? Additionally, is there a change in pathogen levels in the clots as a function of each organ tested post aLy6G or chlodronate cell elimination? These are important questions in order to delineate the proposed mechanism, that at this point, although interesting, is not convincing.
4. The impact of chlodronate on Kufer cell and spleen-resident macrophages needs to be addressed properly to truly delineate the contribution of circulating monocytes vs tissue resident macrophages.
5. The outlined mechanism TNF α /PSGL1 is more associative than being directly attributed to the pathogen and needs more work. TNF α is not the only cytokine during infection.
6. It is unclear how the authors quantified organ thrombosis. The figures or methods provide limited details if this was one organ section/FOV or overall quantitation of what occurs in the 2 organs. It makes figure 2 really hard to understand in addition to the fact that there is lack of statistics, and if the graphs are truly median \pm SEM, then the error bars suggest lack of difference.
7. The rationale of testing leukocyte presence from 18h to 7 days is missing.

Minor comments:

1. It is unclear how the authors performed Dunnett test post Mann-Whitney test as stated in the figures.
2. The figures lack sufficient details and do not stand alone
3. The methods need more detail when it comes to route of administration, pathogen purification, organ cfu measurements and etc

Reviewer #4

(Remarks to the Author)

In this paper authors focus on the time specificity of thrombosis in the liver and spleen during Salmonella infection and find that thrombosis in the spleen is independent of IFN- γ or the platelet C-type lectin-like receptor CLEC-2, despite both molecules being previously identified as key drivers of thrombosis in the liver. Furthermore, the study identifies platelets, monocytes, and neutrophils as core constituents of thrombi in both organs. The research also involves numerous molecules important in inflammation and thrombosis, including TNF α , TF, CD62E, and PSGL-1. Although some links between these coagulation/inflammatory factors and thrombus formation in the inflammatory environment have been established, several issues throughout the manuscript may undermine the conclusion's support.

1. In the description within the manuscript: 'After infection, clusters of monocytic cells, neutrophils and platelets were observed interacting in the splenic vasculature at 24 hours post-infection (Page 7, lines 129-133).' According to Fig 2D, however, the selected platelet marker is CD49b, which is more commonly considered a marker for NK cells in mice (PMID: 35313938). The platelet marker CD41 used in Fig 2B is not continued here. This creates significant confusion regarding the interpretation of the content of the article.
2. The authors state in the abstract, 'Furthermore, we identified platelets, monocytes, and neutrophils as core constituents of thrombi in both organs,' considering this finding as a primary conclusion. The authors believe that the second part of the Results section can support this conclusion. However, in the Results section, Fig 2C indicates a significant increase in neutrophil infiltration over time in the spleen, whereas monocytes do not show a similar trend. Conversely, in the spleen, only monocytes increase significantly over time, while neutrophils do not exhibit a significant trend. Additionally, the figure legend does not specify the number of experimental animals in each group, and there are no statistical annotations, making it difficult to support the conclusion.
3. The authors propose that TNF α drives thrombosis in the spleen and liver after STm infection. Despite support from in vivo experiments and immunofluorescence data, it remains unclear from in vitro experiments whether neutrophils or monocytes can be induced to express more TNF α after Salmonella infection.
4. The same issue arises regarding TNF α -induced endothelial cell expression of CD62E.
5. Although the authors propose the simultaneous presence of platelets, neutrophils, and monocytes in the spleen and liver during infection, the contribution of their coexistence to thrombus formation after infection is not convincingly demonstrated. It is well established that all three components coexist during infection or thrombotic states, but their specific roles in contributing to thrombus formation after infection remain uncertain.
6. In the last two sections of the Results, the authors extensively argue that TNF α produced by monocytes and neutrophils can induce the production of CD62E and tissue factor (TF) by endothelial cells. PSGL-1 is also widely recognized as being involved in thrombus formation. However, the connection between TF and PSGL-1 lacks novelty.

Version 1:

Reviewer comments:

Reviewer #1

(Remarks to the Author)

Perez-Toledo et al has made substantial changes in their revised manuscript. They have carried out more experiments. These experiments have produced not only additional interesting and novel observations but have also helped to clarify some critical issues which were somewhat unclear previously.

As stated in my previous comments, "the most interesting and significant finding in this study is the chronological differences of thrombosis in the liver and spleen". In their original manuscript, the author attributed the chronological differences of thrombosis to organ-specific kinetics without strong supporting data. The additional data from their new experiments have greatly strengthened this conclusion.

The authors have now fully addressed my concerns and the revised manuscript is a significant improvement both in clarity and scientifically.

In summary, the authors of this manuscript describe a study which investigates inflammatory/immune thrombosis induced by Salmonella Typhimurum infection which can potentially be a serious and life-threatening condition. Their novel results will significantly advance the field of immune thrombosis and related field. The conclusions are now well supported by their findings, particularly with the additional observations. The revised manuscript contains adequate methodological details that will allow reproducibility by other investigators. Overall, the revised manuscript is a substantial improvement from the original version and the results are now significantly stronger than before.

Reviewer #2

(Remarks to the Author)

The revised manuscript and comments have addressed all of my previous concerns.

Reviewer #3

(Remarks to the Author)

The authors have addressed most of my concerns and have greatly improved the scientific impact of their manuscript.

Regarding point 5. "The outlined mechanism TNFa/ PSGL1 is more associative than being directly attributed to the pathogen and needs more work. TNFa is not the only cytokine during infection." The cytokine that should be considered here is IL1b that could come even before TNFa by direct interaction of the Salmonella's LPS (that can also be shed) with TLR4 for instance. Thus the mechanism is still only partially explained by TNFa. This should at least be discussed.

Regarding T3SS and the following statement by the authors "In the current study, our focus has been on the contribution of the host rather than the pathogen per se". It is unclear to this reviewer, how the pathogen impact on the host's cells differs from the contribution of the host. There was a study from a few years ago, showing that *Y. pestis*'s T3SS, which is very similar to Salmonella's, is sufficient to dysregulate the platelet thrombotic function. The authors don't need to address the impact of each system component but at least discuss the potential effect of the pathogen beyond the proposed mechanism.

Reviewer #4

(Remarks to the Author)

The author conducted experiments to investigate the mechanisms underlying thrombus formation in the liver and spleen following STm infection. As noted by the author, the recruitment of neutrophils and monocytes by endothelial cells is a phenotype established by previous research. While this study sheds light on the interactions among these three cell types after infection, it still lacks more direct evidence to demonstrate how locally infiltrating inflammatory cells regulate the coagulation system or platelets.

Point-by-point response to reviewers.

Reviewer #1 (Thrombosis immunology)(Remarks to the Author):

This manuscript by Perez-Toledo et al describes an elegant study about inflammatory signatures or factors that drive thrombosis induced by Salmonella Typhimurium (STm) infection in different organs. The authors had previously identified chronological differences between thrombosis in the spleen and in the liver. In this study they found that the clots in both organs contained neutrophils, monocytes, platelets and fibrin. Neutrophils and monocytes were independently required for thrombosis, the former played a more prominent role in the spleen and the latter in the liver. In addition, TNF α , PSGL-1, tissue factor (TF) and thrombin played important roles in thrombosis in both organs as inhibition of TNF α , PSGL-1 and TF suppressed thrombosis. They concluded that STm infection triggered a chain of related inflammatory / prothrombotic events, including TNF α release, up-regulation of endothelial expression of E-selectin, leukocyte infiltration into the organs, increased TF expression and thrombin production, led to thrombosis. They believe that the organ-specific and the broader prothrombotic mechanisms they identified will allow targeted treatments of inflammatory thrombosis induced by systemic infections.

General Comments:

1. Inflammatory/immune multi-organ thrombosis induced by severe viral and bacterial infections is potentially serious. It could lead to multiple organ damage and death. The mechanisms and cellular pathways mediating this type of thrombosis are still not fully elucidated. The novel findings of this elegantly designed study provide further insights into the pathogenesis of infection-induced thrombosis. Their identification of organ-specific cellular processes or pathways may help to explain why thrombosis occurs at a particular time in a specific organ post-infection. Their data may aid development of new, targeted treatments of inflammatory thrombosis resulting from infections.

2. The investigational approach and experimental designs used are appropriate and scientifically sound. The use of unique transgenic mice and specific inhibitory antibodies, add strength to their data. Their conclusions are logical and supported by their findings. The illustration in Figure 8 is helpful to the readers in understanding their proposed model of STm infection-induced thrombosis.

Thank you for these comments.

3. The most interesting and significant finding is the chronological differences of thrombosis in the liver and the spleen. The authors attributed this to differences in organ-specific kinetics without identifying them nor providing reasons how they would account for the chronological differences. Identification of these specific kinetics would greatly strengthen the manuscript.

We have now extended our studies to include more detail on bacterial distribution at different times after infection (added in response to comment 4 below), neutrophils in the liver (comment 6), and the kinetics of expression of TNF α and CD62E in both organs. To examine TNF α and CD62E expression over time, we stained for these molecules in WT spleens from non-immunized mice or mice that were infected with STm for 4, 8 or 18 hours. For the liver we also stained day 7 tissues (data in new Supplementary Figs. 12 and 13 and lines 202-211). No expression of TNF or CD62E was detected in non-infected spleens (representative images shown). TNF α was detectable from 4 hours post-infection in the spleen, with TNF α associated with both F4/80+ and Ly6G+ cells. In the spleen, CD31+ cell-associated CD62E expression was not detected at 4 hours, but was observed from 8 hours post-infection, coinciding with the onset of thrombosis in this organ. In the liver, TNF was not detected in F4/80+ cells until 18 hours post-infection and was still detected at 7 days post-infection. However, minimal CD31+ cell-associated CD62E was detected in the liver at 18 hours post-infection, this was far greater at 7 days post-infection when thrombi are also detected. These analyses suggest that TNF production precedes CD62E expression in the spleen and liver and that CD62E expression is most discernible when thrombi are also present.

4. The authors hypothesize that bacterial infiltration into the tissues is the priming factor for the thrombosis. They also hypothesize that bacteria, residing in the cells or niches near the blood vessels, prime the immune cells and endothelial cells for thrombosis. However, the authors have not shown that the bacteria actually resided in these cells or niches. This evidence would provide confirmation of their hypothesis.

We have now included additional imaging and data assessing the location of bacteria at 4, 8, and 18 hours in the spleen and 1 day and 7 days in the liver (new supplementary Fig. 2, lines 119-124 and 315-317 in the Results and Discussion, respectively). This imaging shows that bacteria mostly associate with cells of the monocytic/macrophage lineage in the splenic red pulp 4, 8 and 18 hours post-infection. In the liver at day 1, the bacteria associate with F4/80 cells. Staining in both organs show that bacteria can be found near thrombi

and vessels when thrombi are detected, but that bacteria can also be found near vessels in the absence of thrombi. As we have previously reported (PMID: 30401709) bacterial detection within thrombi themselves is an infrequent occurrence. This is consistent with a scenario whereby bacteria are required for this process but not sufficient and that other factors contribute, such as TNF production, which can be produced in vitro and in vivo by Ly6C+ cells and Ly6G+ cells (included in revised version as Supplementary Fig. 10, and also included in response to other reviewers' comments).

5. The authors also state that the bacteria were present at similar and near peak levels in the liver and spleen, through the first week of infection, without indicating the location of the bacteria within the organs. Was the distribution of the bacteria evenly spread throughout the organ or was it uneven? However, it is the presence of bacteria at the "priming" sites adjacent to the blood vessels that is most important for induction of thrombosis. Did the bacterial level at the "priming" site(s) peak at a different time in the spleen (day 1) from that in the liver (day 7)? If so, this could explain the chronological differences. Please discuss.

We have addressed this comment in multiple ways. We isolated and dissected individual liver lobes and cultured bacteria from matching segments from individual mice to assess differences in bacterial numbers per gram of tissue. We did not find any bias in bacterial distribution throughout the liver. This was also consistent with what we observed when staining sections for bacteria. What does change is the host cellular milieu that surround the bacteria (presented here and in PMIDs: 32591391 and 26571395). At both day 1 and day 7 bacteria can be detected proximal to the large vessels in the liver. These data comprise part of the response provided in point 4 above.

The picture is more complicated for the spleen because of the division between red and white pulps. In the white pulp few bacteria are detected. Most bacteria are associated with the red pulp and this is consistent throughout the first week of infection. The red pulp is also the most vascularised area of the spleen and where thrombi are detected. Thus, the bacteria are most enriched in sites where thrombi are present. In contrast to the liver (above), there appear to be more bacteria detectable proximal to the vessels, possibly reflecting the quite significant vascular surface area of the spleen. This may contribute to the faster induction of thrombosis in the spleen by day 1 compared to the liver. These data are now presented in the same text and figures as the response to point 4 above.

In our response to points 4 and 5 we have been cautious in our statements because of the intrinsic difficulties in quantifying our data examining distance from vessels and particularly frequency of bacteria at a particular point in an organ. We can say with certainty that these events occur, but not how common they are.

Specific Comments. Major:

6. Page 7 and Figures 2 and 3. Role of neutrophils and monocytes in STm-induced thrombosis. Using cell depletion studies, the authors showed convincingly that both cell types are required for STm-infection induced thrombosis in the liver and the spleen. Peak level of neutrophils in the spleen at 18hour post-infection coincided with the peak level of thrombosis in this organ, and this is consistent with the prominent role of neutrophil in thrombosis induction. However, in the liver, late rise in neutrophil level peaking at day 7 post-infection is surprising. Neutrophils often respond to acute infection very quickly, usually within first few hours, if not in the first day. Furthermore, in figure 2C, neutrophils at its peak in the liver were only about 3% of CD45+ cells. It seems surprising that neutrophil deletion would have such a profound effect on thrombosis, decreasing the thrombi area per section to zero %. Could the authors explain these surprising findings?

We interpret this as there being two aspects to the reviewer's point. The first is about the modest frequency of neutrophils in the infected organs, and the second is why neutrophils are important if present at low proportions. In response to part one, after infection, there is a substantial accumulation of many cell types in the liver. For instance, there is at least a 10-fold increase in leukocytes overall in the liver by day 7 after infection, with substantial increases in monocytic lineage cells amongst these, as well as increases in NK and T cells and other cell types (PMIDs: 32591391 and 26571395), and this accounts for the modest proportional change. In response to the second point, neutrophils in the tissues are likely to be important, but neutrophils in the blood may also contribute to this process since they are also targeted by antibody-targeted depletion (new Fig. 3B, line 150). Therefore, this process may be co-ordinated by cells in tissues and in the blood in a way we do not yet fully understand. Indeed, thrombi stain strongly for neutrophils, and it may be that neutrophils in the blood and tissues contribute to thrombus development. We have raised this possibility in the revised discussion surrounding the model presented in Fig. 8.

The reviewer also suggests there may be an earlier influx of neutrophils into the liver. To address this, we stained for neutrophils in the livers of mice infected for 4, 8, 18 hours, and 7 days after infection. This shows that the reviewer is correct, and there is an earlier accumulation of neutrophils that peaks at 4 hours post-

infection. We have added these data as Supplementary Fig. 5. Therefore, whilst neutrophilia in the liver occurs rapidly after infection, this occurs without thrombosis.

7. It is well known that neutrophils contribute to thrombosis via NETosis. Since the authors showed clear evidence that neutrophils played a critical role in STm infection-induced thrombosis, do they have any data on the contribution of NETosis to STm infection-induced thrombosis?

The reviewer highlights that NETosis may be a contributor to thrombosis here. To examine this in more detail, we first tested in vitro to see if isolated neutrophils could produce NETs upon stimulation with STm. As shown in the results (lines 155-161) and Supplementary Fig. 8, STm stimulation induced NETs production in vitro. Staining of spleen and liver sections for the NETs marker citrullinated histone 3 (CitH3) shows that CitH3 was detected in both spleen and liver thrombi. To examine the role of NETS in thrombosis, we performed experiments using a standard approach previously used by other groups (PMID: 22451716), where we treated mice with DNaseI 6 hours after infection, and then evaluated thrombi in the spleen at 24 hours post-infection. These experiments showed that although treatment with DNaseI reduces the area occupied by thrombi in the spleen, this effect is not seen in every mouse. Amongst the interpretations of these results are that NET formation probably contributes to thrombus formation in the spleen but that this may not be the only mechanism involved.

Minor Comments:

8. The authors need to provide more methodological details about how thrombi areas/section were quantified. Was the counting performed objectively without observer's bias? For example, was counting was done independently by at least two persons? Were the fields of view (VOC) randomly selected to avoid selection bias in the results?

Thrombi areas were quantified using the Zen lite software (Zeiss). The frequency of area occupied by thrombi was quantified by selecting thrombi areas per section (positive staining for both fibrin and CD41), then adding them all up to obtain the total thrombi area, which was then divided by the total area of the section and multiplied by 100 to get a percentage. We did this for every mouse in the experiments and assigned 0 when no thrombi were detected. To quantify other parameters, at least 5 fields of view were selected randomly. For clarity, we have included a diagram in Supplementary Fig. 14 showing this and added a paragraph in the methods section (lines 440-451) describing the image analysis in more detail. The quantification was performed blindly.

9. Antibody staining in some figures is suboptimal. The staining, particularly the blue stain, is so weak that it is very difficult to see. For examples, in Fig 3C, the blue stain in CD41+ platelets is very faint and hence it is very difficult to see the blue-stained platelets in the IC and α Ly6G pictures. The same with the blue fibrin stain in Fig 4D Liver picture, and Fig 6H Liver mTF+/, hTF+/+

We thank the reviewer for this comment. After a close inspection of the mentioned images, we agree that the blue stain in some of them was not very clear, so we have changed the images to improve clarity. We have also included higher-magnification images of the thrombus.

9. On page 10, lines 217-219. The sentence, "In contrast, in the liver day 7 post-infection fewer CD11b+ Ly6G- cells are detected, but frequency and numbers of CD11b+ Ly6C+ cells were similar between the two group (Fig 7F and &7G)". There is a mistake(s) in this sentence. Firstly, Fig 7G shows in day 7 liver picture, CD11b+ Ly6C+ cells are significantly more than in IC mice than in α -PSGL1 treated mice, not similar in numbers as stated in the above sentence. No CD11b+ Ly6G- mice are shown in either group of mice (Fig 7F and &7G). The authors probably mean CD11b+ Ly6G+ mice. If so, Fig 7G shows the number of CD11b+ Ly6G+ cells are the same in the two groups, not fewer as indicated by the authors in the above sentence.

We apologize for any confusion here and have modified the section (lines 244-253) to improve clarity. The reviewer is correct, the frequency of CD11b+Ly6C+ in the liver is reduced after treatment with anti-PSGL-1, but that CD11b+Ly6G+ remains similar. We have also reordered the panels in Fig. 7 to reflect the flow of the results. Furthermore, we also would like to explain that when we referred to CD11b+ Ly6G-, we meant cells that are likely to be monocytes. Since Ly6C can also stain blood vessels, we co-stained for CD11b and Ly6G and considered cells expressing both (CD11b+ Ly6G+) as neutrophils and cells expressing CD11b but not Ly6G (CD11b+ Ly6G-) as monocytes. Thus, using CD11b+Ly6G- is appropriate.

10. Supplementary figure 4. It would be helpful to show data of Wild Type (control) mice for comparison.

We thank the reviewer for this comment. We have now added WT controls to this figure (new Supplementary Figure 11).

Reviewer #2 (Intra-vital Microscopy, bacterial infection)(Remarks to the Author):

In this study, Perez-Toledo and colleagues examine the cellular and molecular mechanisms that drive sTm-driven thrombosis formation, and define differences in these processes in the spleen and liver shortly after infection in mice. While both neutrophils and monocytes are found in thrombi at both sites, their relative contributions seems to differ, suggesting distinct mechanisms. The involvement of TNF α , CD62E and PSGL-1 in this process is demonstrated through blockade and KO studies, and collectively show a step-wise inflammatory cascade that regulate thrombus formation. This study suggests that while the basic composition of thrombi are similar in the spleen and liver, there may be avenues for select therapies that target processes that are specific for a particular organ system. This is an interesting and well written study that builds on previous knowledge and model systems established by this group, and further describes thrombi mechanisms at two different organ sites that has may have implications on disease progression in humans. I have the following comments:

1. Abstract, Line 113: "These results suggest that the mechanism of thrombus formation in the spleen is independent of IFN γ and CLEC-2" is not supported by data in Figure 1B, where % thrombi/section is significantly reduced in IFN γ KO mice. What is the definition of "detected in 50% of mice" (Line 107) and how was this determined? This conclusion needs to be revisited: with CLEC-2 expression being the only significant difference between thrombi formation at the two tissue sites, the weight of this statement and main conclusions of this study becomes less novel. Similarly, in Figure 8: the role of IFN γ is ignored in the spleen. What is the rationale to diminish their contribution here?

We have clarified these points and have changed the text referred to in the results and in the abstract lines 33-34 We have also modified the manuscript to "IFN- γ -deficient mice had reduced thrombosis in the spleen, but thrombi were still detected in 7 out of 10 mice". The percentage figure refers to the percentage of a section that is thrombus (new Fig. 1B, lines 105-106). The finding of thrombosis in IFN- γ -deficient mice is of major significance. Our previous work in the liver showed that thrombosis is completely abrogated in the liver in all mice in the absence of IFN- γ (PMID: 26571395). New data showing the lack of thrombosis in the liver has now been included (new Fig. 1C and 1D, line 107). By finding here that there is only a partial dependence, it indicates that there are distinct features of the mechanisms of thrombosis in the spleen and liver that. This is important because we are observing the development of thrombosis in two different organs, with different kinetics, and potentially driven, at least in part, by different mechanisms. Thank you for pointing out the omission in Fig 8 regarding IFN γ in the spleen, this has now been included.

2. Observations that thrombi formation in spleen after STm infection is unchanged in mice lacking Clec-2 on platelet cells is interesting, but only timepoints up to 24 hours is shown. Can you rule out that the kinetics of thrombus formation is unchanged in the absence of Clec-2 beyond 24 hours during the resolution phase of infection? While thrombus development seems to resolve within a few days in wildtype spleen, it is important to confirm that there is no differences between the groups over time. Do individual thrombus persists throughout the infection or do they form/collapse in waves? Can this data also be presented as % occlusion? *The reviewer's comment contains three different points that we have separated out below.*

- *Are the kinetics of thrombus formation unchanged in the absence of Clec-2 beyond 24 hours during the resolution phase of infection?*

We have looked in the spleen at times up to day 7, when, in this model, the resolution phase of infection typically starts (PMID: 17475847). At day 7, thrombosis has largely resolved in the spleens of both WT and CLEC-2^{fl/fl}PF4^{Cre} mice and has done so to a similar level, such that thrombi are rarely detected in both groups of mice (no mouse we examined had more than 1 thrombus per section at day 7). Therefore, there is no delayed thrombosis in CLEC-2^{fl/fl}PF4^{Cre} mice. Additionally, CLEC-2^{fl/fl}PF4^{Cre} mice clear the infection with kinetics similar to that as WT mice (PMID: 26571395). The few thrombi that are detected look similar in both WT and CLEC-2^{fl/fl}PF4^{Cre} mice. We have included this data as Supplementary Fig. 1C and referred to this data in the manuscript in line 111.

- *Do individual thrombi persist throughout the infection, or do they form/collapse?*

This is a great and fundamental question, but unfortunately, it is technically challenging to answer, and we do not know yet. We aim to assess this as part of future studies.

- *Can this data be presented as % occlusion?*

We have considered this in depth in the past. We concluded that due to the highly vascular nature of the spleen and the difficulty in delineating vessels in the spleen and accurately defining the edge of splenic blood vessels, it is most reproducible and simplest to present thrombosis levels as areas occupied by thrombi

instead.

3. Figure 1B/D, 3A/C, 4E, 5C/D, 6 etc: the figure legend indicates that each data point is from a single mouse. I find it odd that a single image from the spleen is taken from each mouse for quantitation. I suggest the authors analyze multiple images from different splenic sections and plot the mean values from each mice for a more robust, representative data analysis. This is especially important for Figure 1 as it lays the foundation for the main rationale of this study.

We apologise for any lack of clarity here. All the quantifications presented in the current study represent a whole tissue section from the organ. To reassure the reviewer about the reproducibility and representative nature of our results, we assessed how variable thrombosis was within the same organ from four different mice. Our analysis shows that thrombus quantification is consistent for an individual mouse between different tissue sections. In the absence of infection, we do not see thrombi. This is now included in Supplementary Fig 14 and in the image quantification section of the methods. Briefly, whole spleen sections were stained for CD41 and fibrin and scanned with an Axio Scan.Z1 (Zeiss). The frequency of area occupied by thrombi was quantified by selecting thrombi areas per section (positive staining for both fibrin and CD41), then totalling these numbers to obtain the total thrombi area. This number was then divided by the total area of the section and multiplied by 100 to get a percentage. We did this for each mouse in the experiments and assigned 0 when no thrombi were detected.

4. The methods and figure legends do not indicate the sex of mice used in all experiments: please clarify if either or both were used and in which studies stated in the figure legends. If only one sex is used, justification for this is needed and conclusions reflecting this (eg abstract, discussion). It is indicated that IVM studies were done in males: again, a justification is needed why imaging studies were performed exclusively in males. Please also include animal numbers for each figure panel in the legends.

We have now specified in the methods sections that throughout our studies, except for IVM, both female and male mice were used in experiments. We have not observed an obvious difference between male and female mice in our studies. Because we had not demonstrated a significant difference between males and females our subsequent IVM studies we considered it acceptable to only utilize males in these studies.

5. I'm having difficulties interpreting data presented in Figure 2. With both neutrophil and monocytes found in thrombi at both sites, does the increased neutrophil accumulation in spleen (and increase of monocytes in the liver) indicate that there is more of each cell type in each thrombi or are they accumulating in other regions of the organ? To complement the IVM studies, the authors should perform IHC to determine whether discrete regions of the spleen occupy different proportions of the two immune cell types. Can we conclude that increased neutrophil counts in spleen is solely due to increased thrombi formation?

We have added IF of the spleen for day 1 infected mice (Supplementary Fig. 4) to contextualise the localisation of neutrophils, monocytes and platelets together. This shows neutrophils and monocytic-lineage cells are concentrated in the red pulp of the spleen, which is the most vascularised region in the spleen and that the increase in immune cells is not solely due to their presence in thrombi. We have also edited the final sentence of this section for clarity.

6. TNF α blockade has a drastic effect on thrombus formation in vivo, yet its effect on neutrophil homing and monocytes is either modest or non-significant. How do the authors explain the role of these innate cells here? Do platelet numbers in the blood go down after treatment as an alternative explanation? Similarly in Figure 7E: PSGL-1 blockade seems to decrease neutrophil recruitment into the spleen but not monocytes, whereas the reverse is true in the liver. Does this indicate that thrombus formation is driven by different cell types in these organs?

7. There is higher accumulation of neutrophils in the spleen compared to monocytes in Figure 2C, but higher numbers of TNF α + monocytes in both organs. Please discuss this in the context of their respective contribution to thrombus formation.

These are overlapping points and have addressed these together.

After infection, platelet numbers are similar in control mice and after targeting of the TNF pathway.

We agree with the reviewer that these studies are suggesting that there may be a distinct contribution to thrombosis in the spleen and liver by different cell-types. At this stage we do not know all the roles that they may play in both these organs, but it is reasonable to suggest that they contribute TNF α locally, which is a critical driver of this thrombosis cascade. They could also differentially contribute Tissue Factor or promote NET formation in these sites or other effects on the local vasculature

We have now added some discussion on this point (in conjunction with our response to reviewer 4), highlighting that although both cell-types can produce TNF α (new Supplementary Fig. 10 and line 176).

Minor comment:

1. Please include IVM from uninfected spleens for studies in Figure 2D.

This is now added.

2. Line 182: I would argue that CD62E expression facilitates leukocyte rolling and retention, rather than migration.

This has now been changed in the text in line 211.

3. Please replace low resolution images with higher resolution ones: Figure 1A/C, Figure 2D, Fig 7D/F etc.

This is now done

4. Figure 3: some experiments include 4 mice and other show data using 8 mice. For consistency, please include all data in this figure from 2 independent (n=4) data for a total of 8 mice.

We have now done this for the key panels showing thrombosis (Fig. 3D and F). The flow cytometry in Fig. 3A and counts in Fig. 3B were not collected in the same way for both experiments and are only included to indicate that the anti-Ly6G treatment affects cell numbers. Therefore, although experiments had similar results, they cannot be presented as pooled data. These points have been clarified in the legend to Fig. 3.

5. Figure 6A: please use a different color for the median line, it is not visible when everything is in black.

We have changed the colour of the median line and brought it forward to make it visible.

6. Line 318: please check ketamine dose used (likely 200mg/kg)

This has been changed to show the correct dosing and it was 200 mg/kg.

7. Please include higher magnification images for Figure 5A/B, 6D to better visualize co-localization between the indicated signals.

These have been changed

8. What are the effects of anti-platelet agents, and would the authors predict that they will clear thrombi in the spleen and liver equally?

We have only evaluated the effects of anti-platelet agents in the liver after 7 days of infection, and we did not find an effect on preventing thrombosis. We have also only evaluated the prophylactic effect on thrombus formation and not as therapeutic agents. Based on our current understanding of the processes, we do not think these would have an effect, although this would need to be confirmed in future studies.

Reviewer #3 (Infectious thrombosis)(Remarks to the Author):

This manuscript aims to address the contribution of systemic Salmonella infection to differences in innate immune cell contribution to organ thrombosis. Overall, the results of neutrophil monocyte differences in the course of infection-mediated thrombosis in liver vs spleen are interesting and the intravital images and movies are wonderful.

Major concerns:

1. It is unclear to this reviewer why COVID19 in this manuscript is equated to the pathophysiology of Salmonella infection. Do the authors mean to imply that thrombotic outcomes as a function of intestinal bacterial infection and lung viral infection are the same? If so then there needs to be a lot more evidence provided for the reader. I would remove any reference that Salmonella can somehow similarly affect thrombosis as COVID, as it misleads the reader.

We apologize for any confusion that arose from starting with COVID. We intended to use SARS-CoV-2 infection as an example of an infection that induces thrombosis and where thrombosis is associated with poor patient outcomes. To avoid confusion, we have removed reference to SARS-CoV-2 in the manuscript as per reviewer's suggestion.

2. The use of this particular aroA-negative strain of Salmonella is also puzzling when it comes to translational implications. AroA elimination from Salmonella leads to an overall increase of TNF α from the host when compared to the wt. strain in addition to "increased sensitivity to penicillin, complement, and phagocytic

uptake" (PMID:27601574) changing the overall impact and pathophysiology of infection. In addition, the virulence of the strain is attenuated and thus the lack of change in replication in organs is not surprising. Replicating the results with an appropriate virulent (non-modified) Salmonella is of essence for the proper interpretation and generation of these potentially valuable results.

The reviewer has a concern that thrombosis may be a reflection of using an attenuated strain. We can reassure the reviewer that we have previously shown that thrombosis is induced in the spleen by the virulent parent strain (SL1344) of the AroA Salmonella strain (SL3261) used here. In our previous study, infection with similar doses of either SL1344 and AroA SL3261 induced thrombosis in the spleen at day 1 to a comparable level (PMID: 30401709). Our unpublished data shows that the thrombi induced by both the attenuated SL3261 and virulent SL1344 strains in the spleen are indistinguishable histologically including being platelet-rich and having fibrin caps as well as other similarities. Because of mouse genetics the virulent SL1344 strain cannot be studied in BALB/c and C57BL/6 mice (Nramp1 mutants) at day 7, when thrombosis in the liver is detectable, as this bacterial and mouse combination is lethal in mice before this time-point – even when we infect with just 10 SL1344 bacteria. Other groups have also reported thrombosis incidentally within their studies. Bumann and colleagues identified thrombosis in the spleen after infection of BALB/c mice with STm SL1344 (PMID: 34911764). Moreover, after infection of the resistant mouse strain sv129s6 mice, which does not carry the Nramp1 mutation, with SL1344 then thrombosis in the spleen and in the liver can occur (PMID: 20195482). Therefore, thrombosis is not a consequence of infection with AroA-deficient attenuated Salmonella. We have not added the data on SL1344 here because it does not materially change the conclusions of the manuscript.

We have also examined TNF α induction by innate immune cells. To do this we purified and infected mouse monocytes and neutrophils and infected for 2 hours with either SL3261 (AroA) or SL1344 at different multiplicities of infection. No difference was observed in the TNF response induced to either bacterial strain in either monocytes or neutrophils. These results are now part of Supplementary Fig. 10 and referred to in the manuscript in lines 176.

3. Coagulation and immuno-thrombotic formation during bacterial infection serves the purpose of containing and confining the bacteria preventing it from spreading. Does the use of this attenuated strain interfere with the process allowing for the pathogen dissemination to other organs? Does the T3SS play a role for thrombotic outcomes or dissemination? Additionally, is there a change in pathogen levels in the clots as a function of each organ tested post aLy6G or clodronate cell elimination? These are important questions in order to delineate the proposed mechanism, that at this point, although interesting, is not convincing.

We agree that pathogen-containment is one of the common functions proposed for thrombi post-infection. This does not appear to be a function associated with thrombosis after Salmonella infection. We have shown that STm-induced splenic or liver thrombi contain few bacteria (PMID: 30401709) and our unpublished data shows that this is the case in mice infected for 24 hours with the virulent SL1344 strain and is thus not the case for both the virulent and attenuated strains.

The AroA strain does not have an issue in disseminating throughout the mouse and after infection of WT mice is found in sites such as the kidney, lungs, bone marrow and brain as well as the spleen and liver. Of key relevance is that most of the interventions tested in the current study have a minimal impact on the ability to control bacterial numbers despite having a large impact on the level of thrombosis. Thus, we think that thrombosis may not always have a role in containing infection, at least for Salmonella infection. It is not possible to assess whether thrombi after anti-Ly6G or clodronate treatment contain more bacteria since no thrombi form after either of these treatments.

We agree it is important to assess the roles of key bacterial pathways such as the T3SS pathway. In the current study, our focus has been on the contribution of the host rather than the pathogen per se. Further studies would need to include detailed studies on major bacterial pathways and how they contribute. One of the issues with studies using bacteria deficient in SPI2 for instance is that how the bacteria distribute within organs such as the liver. All of our published and unpublished data supports SL3261 AroA and SL1344 having similar tropisms and distributions within host organs. However, SPI2 TTSS mutants such as those lacking ssaV or sseB show a different bacterial distribution within organs such as the liver when compared to WT bacterial control strains (PMID: 23236281). This is why, at this stage, this has not been pursued.

4. The impact of clodronate on Kufer cell and spleen-resident macrophages needs to be addressed properly to truly delineate the contribution of circulating monocytes vs tissue resident macrophages.

We agree with the reviewer that further work needs to be done to fully delineate the specific role of different monocytic-lineage subsets in their contribution to thrombosis. We chose to use clodronate since it is a widely used method to deplete cells from the monocytic lineage. Thus, we referred to "monocytic cells" or

"monocytic-lineage cells" in the manuscript to encompass the spectrum of cells that belong to this categorization.

To address the reviewer's question, we performed flow cytometry to test the effect of clodronate on monocytic cells in the spleen and found that clodronate reduced the number of red pulp macrophages (F4/80+CD11b-), but it also reduced the number of Ly6C+ monocytes (F4/80+CD11b+Ly6G-Ly6C+). We also performed immunofluorescence on spleen sections to evaluate the effect of clodronate on other tissue-resident macrophages. We found that clodronate also impacted marginal-zone macrophages (CD169+), whereas metallophilic macrophages (SIGN-R1+) remained present. In the liver, we assessed the expression of Clec4F, a molecule described as a specific marker for murine Kupffer cells (PMID: 26813785). As expected, Clec4F staining was clearly detectable in the livers of non-immunized mice and of control liposome-treated and infected mice, but not in mice clodronate-treated and infected. These results are now presented as part of Supplementary Fig. 9 and lines 164-167.

5. The outlined mechanism TNF α / PSGL1 is more associative than being directly attributed to the pathogen and needs more work. TNF α is not the only cytokine during infection.

As the reviewer suggests, infections will trigger an inflammatory response that involves a myriad of cytokines or other soluble factors. If the reviewer is asking whether all major, early cytokines contribute then this is obviously a large additional study in itself. However, we can say that not all key cytokines are important as for instance IL-6 is a central cytokine in the acute phase, but when we examined thrombosis in IL-6KO mice, we found that thrombosis occurs normally in these mice. Thus, thrombosis is not a general response to all cytokines, but a highly regulated specific response, in this instance involving TNF.

6. It is unclear how the authors quantified organ thrombosis. The figures or methods provide limited details if this was one organ section/FOV or overall quantitation of what occurs in the 2 organs. It makes figure 2 really hard to understand in addition to the fact that there is lack of statistics, and if the graphs are truly median +/-SEM, then the error bars suggest lack of difference.

We have now included more details on the methods used. Within the methods (lines 440-451) we have provided more detail on how thrombosis was quantified, as well as a diagram in the supplementary figures (Supplementary Fig. 14) that outlines the process. Briefly, we used the Zen lite software (Zeiss) to determine the percentage of the area of a section occupied by thrombi, with thrombi identified by positive staining for both fibrin and CD41. These areas were then totalled to obtain the total thrombi area, which was then divided by the total area of the section and multiplied by 100 to get a percentage. We did this for every mouse in the experiments and assigned 0 when no thrombi were detected. We have also included the missing statistics in Figure 2.

7. The rationale of testing leukocyte presence from 18h to 7 days is missing.

We have chosen the times based on our previous work (e.g. PMID: 26571395) which identified these time-points as critical periods for detecting thrombosis. We have tried, within reason, to provide relevant matching times between the spleen and liver to simplify making comparisons between these two sites. With thrombosis detected in the spleen shortly after infection (by 24 hours) it is rational to also look at 18 hours, whereas in the liver, where thrombi are detected in the liver at day 7, such a comparison is of more limited value.

Minor comments:

1. It is unclear how the authors performed Dunnett test post Mann-Whitney test as stated in the figures. *Thank you for pointing this out. It was a mistake and has been altered*

2. The figures lack sufficient details and do not stand alone

3. The methods need more detail when it comes to route of administration, pathogen purification, organ cfu measurements and etc

We have now included more detail as requested and edited legends where appropriate to enhance clarity.

Reviewer #4 (Infectious thrombosis)(Remarks to the Author):

In this paper authors focus on the time specificity of thrombosis in the liver and spleen during Salmonella infection and find that thrombosis in the spleen is independent of IFN- γ or the platelet C-type lectin-like receptor CLEC-2, despite both molecules being previously identified as key drivers of thrombosis in the liver. Furthermore, the study identifies platelets, monocytes, and neutrophils as core constituents of thrombi in both organs. The research also involves numerous molecules important in inflammation and thrombosis, including TNF α , TF, CD62E, and PSGL-1. Although some links between these coagulation/inflammatory factors and thrombus formation in the inflammatory environment have been established, several issues throughout the manuscript may undermine the conclusion's support.

1. In the description within the manuscript: 'After infection, clusters of monocytic cells, neutrophils and

platelets were observed interacting in the splenic vasculature at 24 hours post-infection (Page 7, lines 129-133).' According to Fig 2D, however, the selected platelet marker is CD49b, which is more commonly considered a marker for NK cells in mice (PMID: 35313938) . The platelet marker CD41 used in Fig 2B is not continued here. This creates significant confusion regarding the interpretation of the content of the article. *We thank the reviewer for their comment. Indeed, CD49b is also a marker for NK cells. However, the advantage of using this marker over CD41 for intra-vital imaging has been reported before (PMID: 21949865). It has been reported that the in vivo use of the anti-mouse CD41 antibody clone MWR30 can provoke adverse reactions in mice (PMID: 10397735). We also performed experiments using an anti-GPIb antibody, which was used to corroborate the interactions between neutrophils, monocytic cells and platelets (Supplementary video 5). Critically, we have not relied on CD49b in our study to define platelets as we have stained using multiple other markers on IHC/IF that are specific to platelets, such as CD41.*

2. The authors state in the abstract, 'Furthermore, we identified platelets, monocytes, and neutrophils as core constituents of thrombi in both organs,' considering this finding as a primary conclusion. The authors believe that the second part of the Results section can support this conclusion. However, in the Results section, Fig 2C indicates a significant increase in neutrophil infiltration over time in the spleen, whereas monocytes do not show a similar trend. Conversely, in the spleen, only monocytes increase significantly over time, while neutrophils do not exhibit a significant trend.

As the reviewer states, the frequency of neutrophils in the spleen and liver increased after STm infection, and this increase is significantly different at 18 hours post-infection. We also agree that the trend in increasing monocytes is less pronounced. However, assessment of Ly6G and Ly6C staining within CD41+ thrombi shows that neutrophils and monocytes are a consistent feature of all thrombi analysed in both the spleen and liver (newly quantified as the area covered by either Ly6G+ or Ly6C+ across 10-12 different spleen and liver thrombi, consistent across multiple experiments, and included in the revised figures as new Fig. 2C).

Additionally, the figure legend does not specify the number of experimental animals in each group, and there are no statistical annotations, making it difficult to support the conclusion.

We have now expanded the figures and figure legends to provide more detail and now included statistical significance and the number of mice for the experiments.

3. The authors propose that TNF α drives thrombosis in the spleen and liver after STm infection. Despite support from in vivo experiments and immunofluorescence data, it remains unclear from in vitro experiments whether neutrophils or monocytes can be induced to express more TNF α after Salmonella infection.

To assess if these cell-types produce TNF α in vitro we have now carried out experiments where monocytes or neutrophils were purified from bone marrows of naive wild-type mice (C57BL/6) and infected for two hours. We infected with two different strains of STm: SL1344, a wild-type strain, and SL3261, the attenuated strain used in this study for in vivo experiments. We then assessed intracellular TNF production in monocytes (CD11b+Ly6C+) and neutrophils (CD11b+Ly6G+). STm induced TNF in both monocytes and neutrophils. SL1344 and SL3261 induced similar responses at comparable multiplicity of infections. Thus, STm also induces the production of TNF α by monocytes and neutrophils in vitro. These results are now part of Supplementary Figure 10 and line 176 in the manuscript.

4. The same issue arises regarding TNF α -induced endothelial cell expression of CD62E.

The reviewer is correct that we have not shown that TNF α induces CD62E expression in endothelial cells in vitro. This is because of the already extensive evidence base for this within the literature. The discovery of CD62E was made using human endothelial cells that were stimulated with IL-1 β or TNF α (PMID: 3485132, PMID: 2827173). Since then, there has been ample description of the kinetics of induction of CD62E by TNF (PMID: 1713680, PMID: 1697308). Later, it was described that both CD62P and CD62E were inducible in mouse endothelial cell lines in vitro after stimulation with TNF α (PMID: 1378846). Furthermore, the monoclonal antibody used in our study was generated by using a mouse brain capillary endothelioma stimulated with TNF α . Alongside the technical difficulty of isolating murine endothelial cells, and the lack of evidence within our study to challenge the existing knowledge base, we have not performed these experiments. We have strengthened the text to emphasise that CD62E induction in endothelial cells by TNF α is established within the literature.

5. Although the authors propose the simultaneous presence of platelets, neutrophils, and monocytes in the spleen and liver during infection, the contribution of their coexistence to thrombus formation after infection is not convincingly demonstrated. It is well established that all three components coexist during infection or thrombotic states, but their specific roles in contributing to thrombus formation after infection remain uncertain.

The reviewer highlights the lack of knowledge in how cells come together to promote thrombosis. We agree that this is a deficiency in our knowledge. Here we have shown some of the mechanisms associated with multi-organ thrombosis triggered by infection. We acknowledge the reviewer's point that the specific roles

contributing to thrombus formation after infection remain uncertain. However, we consider this to be a work in progress and something that we intend to explore in the future.

6. In the last two sections of the Results, the authors extensively argue that TNF α produced by monocytes and neutrophils can induce the production of CD62E and tissue factor (TF) by endothelial cells. PSGL-1 is also widely recognized as being involved in thrombus formation. However, the connection between TF and PSGL-1 lacks novelty.

We respectfully disagree with the reviewer. In the last 3 sections of our manuscript, we show the roles of TNF, TF, and PSGL-1 in thrombus formation in the spleen and liver. We observed a decrease in CD62E expression in the endothelium in mice treated with an anti-TNF antibody. These results suggest that TNF induction of CD62E in the endothelium facilitates the recruitment of leukocytes and subsequent thrombosis. In contrast, we did not observe any TF staining associated with endothelial cells, suggesting that leukocytes were the primary source of TF that drove thrombosis. Inhibition of PSGL-1 also reduced the recruitment of leukocytes and thrombosis.

Response to reviewers

Reviewer #1 (Remarks to the Author):

Perez-Toledo et al has made substantial changes in their revised manuscript. They have carried out more experiments. These experiments have produced not only additional interesting and novel observations but have also helped to clarify some critical issues which were somewhat unclear previously.

As stated in my previous comments, "the most interesting and significant finding in this study is the chronological differences of thrombosis in the liver and spleen". In their original manuscript, the author attributed the chronological differences of thrombosis to organ-specific kinetics without strong supporting data. The additional data from their new experiments have greatly strengthened this conclusion.

The authors have now fully addressed my concerns and the revised manuscript is a significant improvement both in clarity and scientifically.

In summary, the authors of this manuscript describe a study which investigates inflammatory/immune thrombosis induced by *Salmonella Typhimurum* infection which can potentially be a serious and life-threatening condition. Their novel results will significantly advance the field of immune thrombosis and related field. The conclusions are now well supported by their findings, particularly with the additional observations. The revised manuscript contains adequate methodological details that will allow reproducibility by other investigators. Overall, the revised manuscript is a substantial improvement from the original version and the results are now significantly stronger than before.

We thank the reviewer for their comments and suggestions that have all helped to improve this paper.

Reviewer #2 (Remarks to the Author):

The revised manuscript and comments have addressed all of my previous concerns.

Thank you.

Reviewer #3 (Remarks to the Author):

The authors have addressed most of my concerns and have greatly improved the scientific impact of their manuscript.

Regarding point 5. "The outlined mechanism TNFa/ PSGL1 is more associative than being directly attributed to the pathogen and needs more work. TNFa is not the only cytokine during infection." The cytokine that should be considered here is IL1b that could come even before TNFa by direct interaction of the *Salmonella*'s LPS (that can also be shed) with TLR4 for instance. Thus the mechanism is still only partially explained by TNFa. This should at least be discussed.

We agree with the reviewer that other cytokines are likely involved in mediating thrombosis after STm infection and it is important to use studies such as ours as a springboard for deeper investigations. We thank the reviewer for the suggestion to examine cytokines like IL1 β and we will incorporate this as part of future studies, alongside assessments of pathways such as the inflammasome pathways more broadly. The assessment of TLR4 was identified in our original paper on this subject (JR Hitchcock et al JCI 2015).

We have expanded the discussion to include the need to investigate more cytokines and the possible role of IL-1b in lines 310-312 in the discussion.

Regarding T3SS and the following statement by the authors "In the current study, our focus has been on the contribution of the host rather than the pathogen per se". It is unclear to this reviewer, how the pathogen impact on the host's cells differs from the contribution of the host. There was a study from a few years ago, showing that *Y. pestis*'s T3SS, which is very similar to *Salmonella*'s, is sufficient to dysregulate the platelet thrombotic function. The authors don't need to address the impact of each system component but at least discuss the potential effect of the pathogen beyond the proposed mechanism.

We thank the reviewer for pointing out the manuscript. We have discussed in lines 323-327.

Reviewer #4 (Remarks to the Author):

The author conducted experiments to investigate the mechanisms underlying thrombus formation in the liver and spleen following STM infection. As noted by the author, the recruitment of neutrophils and monocytes by endothelial cells is a phenotype established by previous research. While this study sheds light on the interactions among these three cell types after infection, it still lacks more direct evidence to demonstrate how locally infiltrating inflammatory cells regulate the coagulation system or platelets.

The types of studies the reviewer suggests will be very useful going forward, as we agree that further investigation is required. The scale of such studies will necessitate that they form a stand-alone project.